# Structural basis for the oligomerization-facilitated NLRP3 activation

Xiaodi Yu [1,6] ✉, Rosalie E. Matico[1,6], Robyn Miller[1], Dhruv Chauhan[2], Bertrand Van Schoubroeck[2], Karolien Grauwen[2], Javier Suarez[1], Beth Pietrak[1], Nandan Haloi [3], Yanting Yin[1], Gary John Tresadern [4], Laura Perez-Benito[4], Erik Lindahl [5], Astrid Bottelbergs[4], Daniel Oehlrich[4], Nina Van Opdenbosch[2] & Sujata Sharma [1]

The NACHT-, leucine-rich-repeat-, and pyrin domain-containing protein 3 (NLRP3) is a critical intracellular inflammasome sensor and an important clinical target against inflammation-driven human diseases. Recent studies have elucidated its transition from a closed cage to an activated disk-like inflammasome, but the intermediate activation mechanism remains elusive. Here we report the cryo-electron microscopy structure of NLRP3, which forms an open octamer and undergoes a ~90° hinge rotation at the NACHT domain. Mutations on open octamer's interfaces reduce IL-1β signaling, highlighting its essential role in NLRP3 activation/inflammasome assembly. The centrosomal NIMA-related kinase 7 (NEK7) disrupts large NLRP3 oligomers and forms NEK7/NLRP3 monomers/dimers which is a critical step preceding the assembly of the disk-like inflammasome. These data demonstrate an oligomeric cooperative activation of NLRP3 and provide insight into its inflammasome assembly mechanism.

NLRP3, an extensively studied inflammasome[1–7], belongs to the family of the nucleotide-binding domain (NBD) and leucine-rich repeat (LRR)-containing proteins (NLRs), a crucial component of the cytosolic immunosurveillance system of mammals, detecting signature components of pathogens and consequently triggering immune responses[8–11]. NLRP3 comprises an N-terminal pyrin domain (PYD); a central NACHT; and a C-terminal LRR (Fig. 1A)[12]. In response to cellular stress or damage-associated molecular patterns, activation of NLRP3 is concomitant with the priming step involving the transcriptional upregulation of NLRP3 and pro-inflammatory cytokines, followed by the ATP hydrolysis-induced conformational rearrangement of the NACHT subdomains nucleotide-binding domain (NBD), helical domain 1 (HD1), winged helix domain (WHD), and helical domain 2 (HD2), and interaction with the mitotic serine/threonine kinase NEK7[13–16]. Once activated, the NLRP3 inflammasome recruits the adapter protein ASC,

then activates caspase-1, which in turn cleaves gasdermin D (GSDMD) and pro-inflammatory cytokines pro-IL-1β and pro-IL-18 into their mature forms, resulting in the cellular release of IL-1β and IL-18 through GSDMD pores. This pathway plays an important role in immune responses, but excessive or dysregulated NLRP3 activation and various gain-of-function (GOF) mutations have been linked to numerous inflammatory diseases, such as gout, type 2 diabetes, and neurodegenerative disorders[9,17–21]. Multiple NLRP3 inhibitors have been reported that impair NLRP3 ATP hydrolysis activity, among which MCC950 has been most studied[21,22]. Structures of NLRP3 with MCC950 confirmed it binds in the NACHT domain, in a location distinct from the nucleotide-binding site, and locks NLRP3 in a closed conformation[3–6]. Recently, the disk-shaped structure of fully activated NLRP3 oligomers in complex with ATPγS, NEK7, ASC, and caspase-1 was reported[23]. The structural basis for NLRP3 activation, however, has

[1]Johnson & Johnson Innovation Medicine, Spring House, PA 19044, USA. [2]Johnson & Johnson Innovation Medicine, J&J Interventional Oncology, Beerse, Belgium. [3]Department of Applied Physics, Swedish e-Science Research Center, KTH Royal Institute of Technology, Stockholm, Sweden. [4]Johnson & Johnson Innovation Medicine, Discovery Sciences, Beerse, Belgium. [5]Department of Biochemistry and Biophysics, Science for Life Laboratory, Stockholm University, Stockholm, Sweden. [6]These authors contributed equally: Xiaodi Yu, Rosalie E. Matico. ✉e-mail: xyu6@its.jnj.com

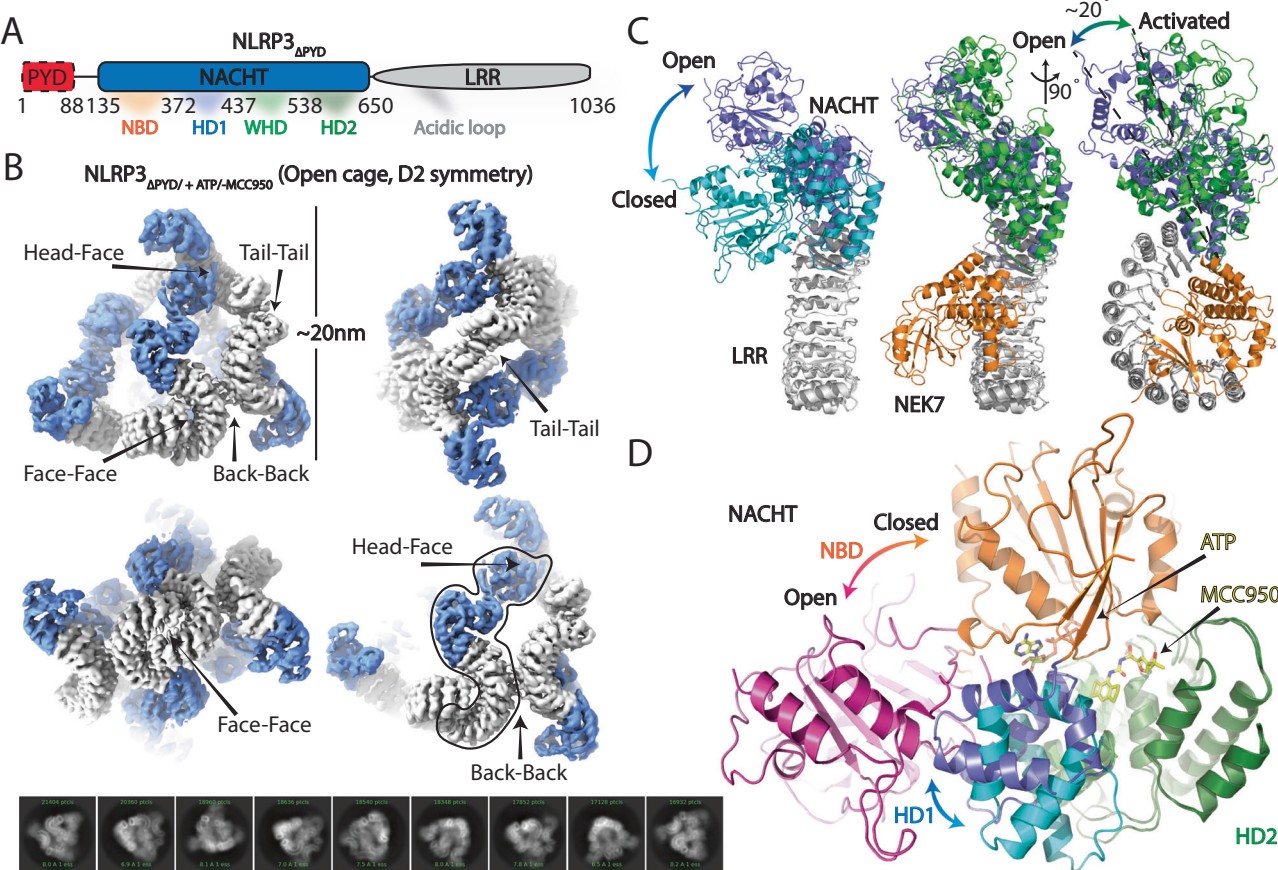

**Fig. 1 | Cryo-EM structure of open NLRP3$_{\Delta PYD/+ATP/-MCC950}$ octamer. A** Domain organization of human NLRP3 with PYD domain truncated. **B** Different views of cryo-EM map of NLRP3$_{\Delta PYD/+ATP/-MCC950}$. The NACHT and LRR domains were in slate and gray, respectively, with the key interfaces between protomers highlighted. One NLRP$_{\Delta PYD}$ protomer was highlighted. 2D classes from cryo-EM images of the sample were shown at the bottom. **C** Structural overlay of open NLRP3$_{\Delta PYD/+ATP/-MCC950}$ to the closed NLRP3 (Left, PDB ID: 7LFH) or activated NEK7/NLRP3 (Right, PDB ID: 8EJ4). The LRR region and NEK7 were colored in gray and orange, respectively. The NACHT domains were in cyan, slate, or green for the closed, open, or activated states, respectively. **D** Zoom-in views of NACHT conformational changes. NBD and HD1 were in orange and cyan or magenta and slate for the closed or open states, respectively. WHD and HD2 were light and dark green, respectively. The nucleotide and MCC950 from the closed NLRP3 were shown as sticks. The protein structures were shown as cartoon. Subdomain reorganization upon activation was highlighted by arrows.

so far remained unknown. In this study, we determine the cryo-electron microscopy (cryo-EM) structures of human NLRP3 in the open oligomer, closed hexamer, and closed NEK7/NLRP3 dimer forms. Together with cellular assays, and molecular dynamic simulations, our findings elucidate a mechanistic model for the oligomeric cooperative activation of the NLRP3 inflammasome.

## Results

### Structure of the open NLRP3$_{\Delta PYD}$ oligomer

PYD-deleted human NLRP3 (NLRP3$_{\Delta PYD}$) was expressed and purified using a baculoviral expression system (Supplementary Fig. 1A). Nano differential scanning fluorometry (nanoDSF) indicated that the presence of ATP and MCC950 synergistically stabilized the purified NLRP3$_{\Delta PYD/+ATP/+MCC950}$ by 0.56°C (Supplementary Fig. 1B). MCC950 inhibits the ATPase hydrolysis activity of NLRP3$_{\Delta PYD}$ with the IC$_{50}$ ~ 128 nM (Supplementary Fig. 1C), confirming inhibition through the expected mechanism[7,22,24,25].

The purified NLRP3$_{\Delta PYD}$ with ATP, but in the absence of MCC950, resulted in non-productive aggregate formation within 30 mins after size exclusion chromatography (SEC). In order to assess the structural state of NLRP3$_{\Delta PYD/+ATP/-MCC950}$, the purified complex was applied to the EM grids immediately after SEC fractionation. Initial 2D classification showed significant heterogeneity, with both small and large complex particles (Supplementary Fig. 1D). One stable class of

particles was isolated through multiple rounds of 2D and 3D classification, and further refined by imposing D2 symmetry (Fig. 1B, Supplementary Table 1, Supplementary Figs. 2, 3). The NLRP3$_{\Delta PYD/+ATP/-MCC950}$ complex was modeled as an octamer with 8 identical NLRP3$_{\Delta PYD/+ATP/-MCC950}$ protomers (Fig. 1B, Supplementary Movie 1). A structural overlay of NLRP3$_{\Delta PYD/+ATP/-MCC950}$ and the closed cage (PDB ID: 7PZC) protomers using LRR as the reference revealed large conformational changes in the NACHT domain (Fig. 1C left). The NBD flips up using HD1 as the pivot point (Fig. 1D). The MCC950 binding site in the closed cage is strikingly different in the NLRP3$_{\Delta PYD/+ATP/-MCC950}$ octamer, where it is only partially formed due to large conformational changes in NACHT (Fig. 1D). In addition, the NLRP3$_{\Delta PYD/+ATP/-MCC950}$ and activated human NLRP3 inflammasome protomers (PDB ID: 8EJ4) can be overlayed with the root-mean-square deviation (RMSD) 8.3 Å (Fig. 1C). Superposition, using LRR as the reference, reveals small variations of the NLRP3$_{\Delta PYD/+ATP/-MCC950}$ protomer compared with the activated protomer, whilst a ~20° rigid body rotation of NACHT is needed to fully overlay the octamer and the activated inflammasome protomers. Superpositions of equivalent subdomains (NBD, HD1, WHD, HD2, and LRR) from the closed, open, and activated NLRP3 structures further improve the structural similarities, highlighting the NLRP3's plasticity with predominant subdomain rigid body movements during conformational changes (Supplementary Fig. 4, and Supplementary Table 2). Taken together, these

observations indicated that NLRP3$_{\Delta PYD/+ATP/-MCC950}$ adopts an open pose in the octamer.

## The interfaces of open NLRP3$_{\Delta PYD}$ within the oligomer

The open form NLRP3$_{\Delta PYD/+ATP/-MCC950}$ octamer formed through Back-Back (LRR-LRR) interaction, and united via the Head-Face (NACHT-LRR), Tail-Tail (LRR-LRR), and Face-Face (LRR-LRR) interactions (Fig. 2A). The Back-Back interaction is conserved in the closed full-length (FL) NLRP3 cages[4–6], closed NLRP3$_{\Delta PYD}$ hexamer[4], and open NLRP3$_{\Delta PYD/+ATP/-MCC950}$ octamer complexes and are predominantly hydrophobic (Fig. 2B). Two C2 symmetry-related salt-bridge pairs were observed between Asp$_{789}$ and Arg$_{816}$ from individual NLRP3 protomers. A Head-Face interaction was identified, which occurs between the positively charged surface of the NBD domain and the negatively charged surface on the concave side of the neighboring LRR (Fig. 2A, Supplementary Fig. 5). NBD- Lys$_{133}$, Arg$_{137}$, Lys$_{138}$, Lys$_{142}$, Lys$_{289}$ can potentially form salts-bridges with corresponding LRR- Asp$_{804}$, Asp$_{807}$, Glu$_{864}$, Glu$_{1007}$, Asp$_{750}$, respectively (Fig. 2C, Supplementary Fig. 3B). The octamer structure displayed two pairs of Tail-Tail interfaces, which involve hydrophobic interactions located at the distal ends of the LRRs. In particular, LRR- Leu$_{984}$, Met$_{987}$, Met$_{988}$, and Trp$_{959}$ forms a small hydrophobic patch that interacts with the neighboring LRR (Fig. 2A, D). In addition, two pairs of Face-Face interactions with

extensive complementary charge-based interfaces were formed between LRRs at the concave side (Fig. 2E, Supplementary Fig. 6). Similar Face-Face interfaces were also observed in closed human and mouse FL NLRP3 cages[5,6] (Fig. 2E). In the closed human FL NLRP3 cage (PDB ID: 7PZC), acidic loops are situated on the concave side of the LRR, facilitating Face-Face interactions. In contrast, these loops are disordered in the closed mouse FL NLRP3 cage (PDB ID: 7LFH) and the open human NLRP3 octamer. The acidic loops induce ~20° twist in the LRRs when comparing the human and mouse Face-Face interfaces in the closed cages, and a roughly 60° twist in the LRRs at the Face-Face interfaces between the human closed cage and open octamer (Supplementary Fig. 6A). In addition, the Face-Face interactions were missing from the closed NLRP3$_{\Delta PYD}$ hexamer structures (PDB ID: 7VTP, 7ZGU)[4]. These observations suggest flexibility in the Face-Face interactions within the closed and open oligomerization states (Supplementary Fig. 6A, B), allowing cage structural rearrangement.

## Proposed NLRP3 activation mechanism

Next, we would like to explore if the open NLRP3 octamer can be derived from the closed cage complex. The closed FL NLRP3 cages formed through repetitive Back-Back and Face-Face interactions among the LRRs[4–6]. For simplicity in the following discussion, we'll refer to the NLRP3 dimers with Back-Back interactions as one unit,

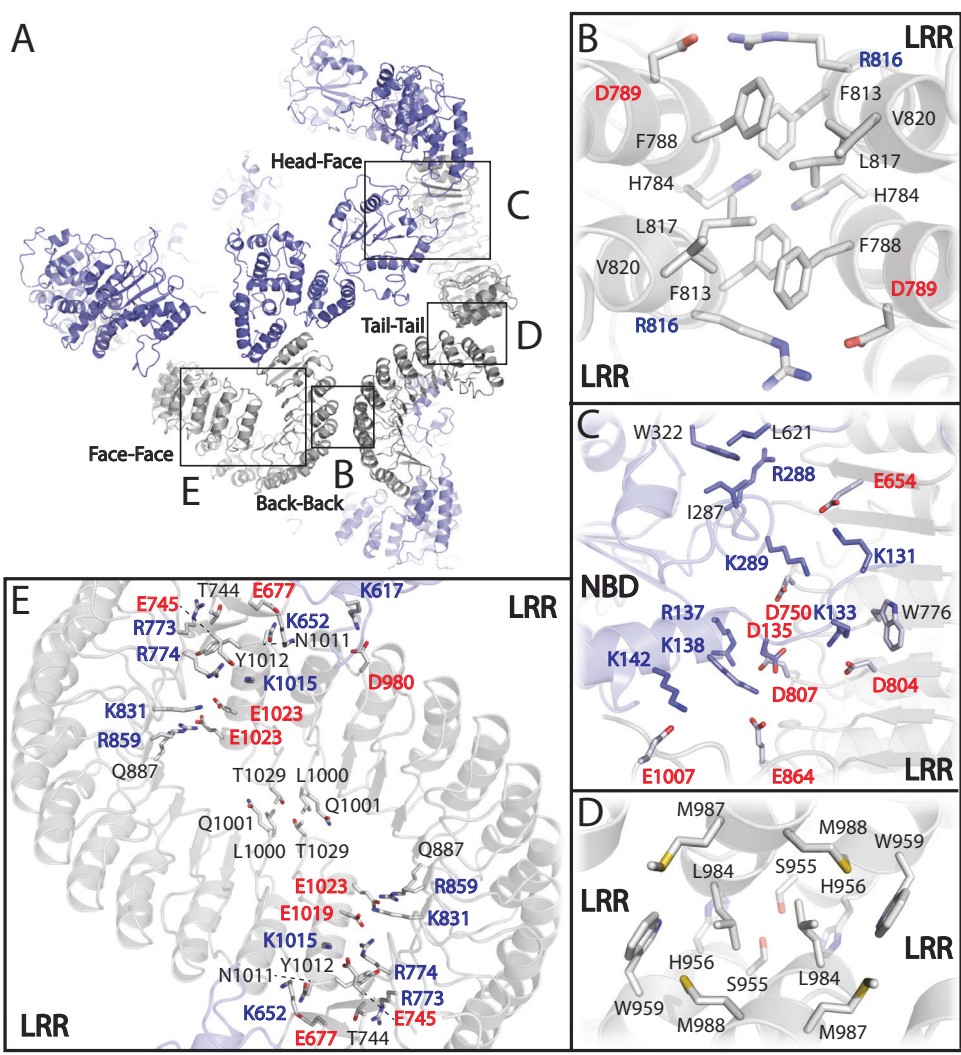

**Fig. 2 | Interfaces in NLRP3$_{\Delta PYD/+ATP/-MCC950}$ octamer assembly. A** Interfaces in NLRP3$_{\Delta PYD/+ATP/-MCC950}$ octamer assembly. Squares indicated the locations of the interfaces. **B–E** Zoomed-in views of Back-Back, Head-Face, Tail-Tail, or Face-Face interfaces, respectively. The main and side chains of residues at the interfaces were shown as cartoon and sticks, respectively, and following the color code in Fig. 1A.

and four consecutive units with Face-Face interactions were used to elucidate the activation mechanism (Fig. 3A). Compared to the closed cages, the open NLRP3$_{\Delta PYD}$ octamer features additional Head-Face and Tail-Tail interactions. A structural overlay using one unit (Fig. 3B, C, 1 (1') and 2 (2') protomers) as the reference reveals that the Face-Face interactions at the concave sides of LRRs in the open octamer are strikingly different from those in the closed cage (Fig. 3A−C, between 2 (2') and 3 (3') protomers). Looking from the reference planar, the Face-Face interaction in the open octamer is rotated anticlockwise by ~40 degrees relative to that in the closed cage (Fig. 3B, C). Together with another C2 symmetry-related Face-Face interaction in the open octamer, the anticlockwise rotations at both Face-Face interactions create additional Tail-Tail interactions between the catalytic LRRs (Fig. 3B, C, between 4 and 4' protomers). The formation of Tail-Tail interactions further exposes a large negatively charged catalytic surface at the LRR from the 4, or 4' NLRP3, which attracts the positively charged receptor interface at the NBD from the 3' or 3 NLRP3, respectively (Fig. 3D, Supplementary Fig. 5, Supplementary Movie 2).

To assess the impact of PYD domains on the open NLRP3 octamer, we conducted molecular dynamic (MD) simulations using a chimera model derived from the open NLRP3$_{\Delta PYD}$ structure and the PYD subdomain model from the AlphaFold2[26] (AF2) FL NLRP3 model (AF-Q96P20-F1). The PYD$_{1-147}$ (amino acid (a.a.), 1–147) model was isolated from the human FL NLRP3 AF2 model and fused to each individual

protomer within the open octamer using the α-helix$_{136-147}$ as the reference. The resulting chimera FL NLRP3 open octamer was subjected to the default relaxation protocol in Maestro and followed by a 100 ns MD simulation (Supplementary Movie 3). The FL NLRP3 open octamer model maintained stability throughout the MD simulation, with consistent Back-Back, Head-Face, Face-Face, and Tail-Tail interactions. Notably, the modeled PYDs exhibited relatively stable interactions at the NBD's rear, forming Head-Face interactions (Supplementary Fig. 7A). This putative PYD docking site was exclusively present in the open and activated NLRP3 form, partially obscured by the WHD domain in the closed conformation (Supplementary Fig. 7B). At the edge of catalytic LRR, one positively charged patch could serve as an extra docking site for the predominantly negatively charged α-helix$_{114-128}$ from the receptor NLRP3 protomer (Supplementary Fig. 7C), which can assist the conformational changes on the NACHT, meanwhile positioning the PYD domain.

## Oligomeric interfaces are critical for NLRP3 activation and inflammasome assembly

To assess the physiological significance of the open NLRP3$_{\Delta PYD/+ATP/-MCC950}$ structure and its octamer state, multiple single mutations were introduced to the FL NLRP3 at the oligomerization interfaces and examined in a cellular assay (Fig. 4A, Supplementary Fig. 8). To mimic physiological NLRP3 concentrations after priming in various cell types, different concentrations (2.5 ng, 5 ng, and 10 ng) of NLRP3 vectors

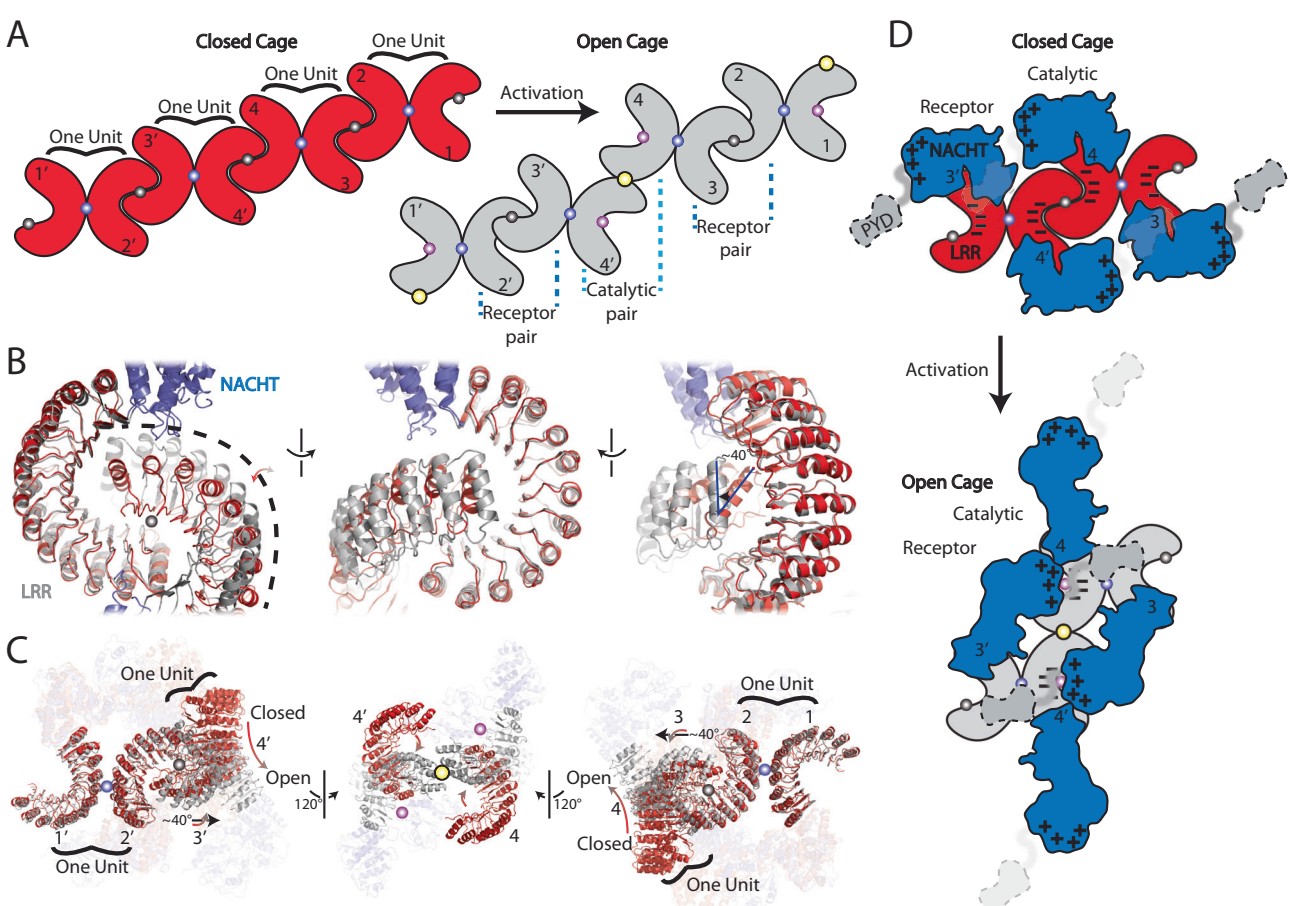

**Fig. 3 | Purposed NLRP3 activation mechanism. A** Schematic representation of the arrangement of the oligomerization transition from the closed to the open cages. NACHT was not shown. The Back-Back, Face-Face, Head-Face, and Tail-Tail interactions were highlighted using magenta, gray, pink, and yellow spheres, respectively. NLRP3 dimers with Back-Back interactions was highlighted as one unit. **B** Structural overlays of NLRP3 closed cage (PDB ID: 7LFH) and open octamer

(this study) at face-face interface using one protomer as the reference. NACHT were colored in slate. LRR were in red or gray for closed cage or open octamer, respectively. **C** Structural overlays of NLRP3 closed (red, PDB ID: 7LFH) and open (slate and gray, this study) cages. LRRs were highlighted. **D** Schematic representation of the proposed activation mechanism at the NACHT domain in the octamer. Note, two consecutive units were shown.

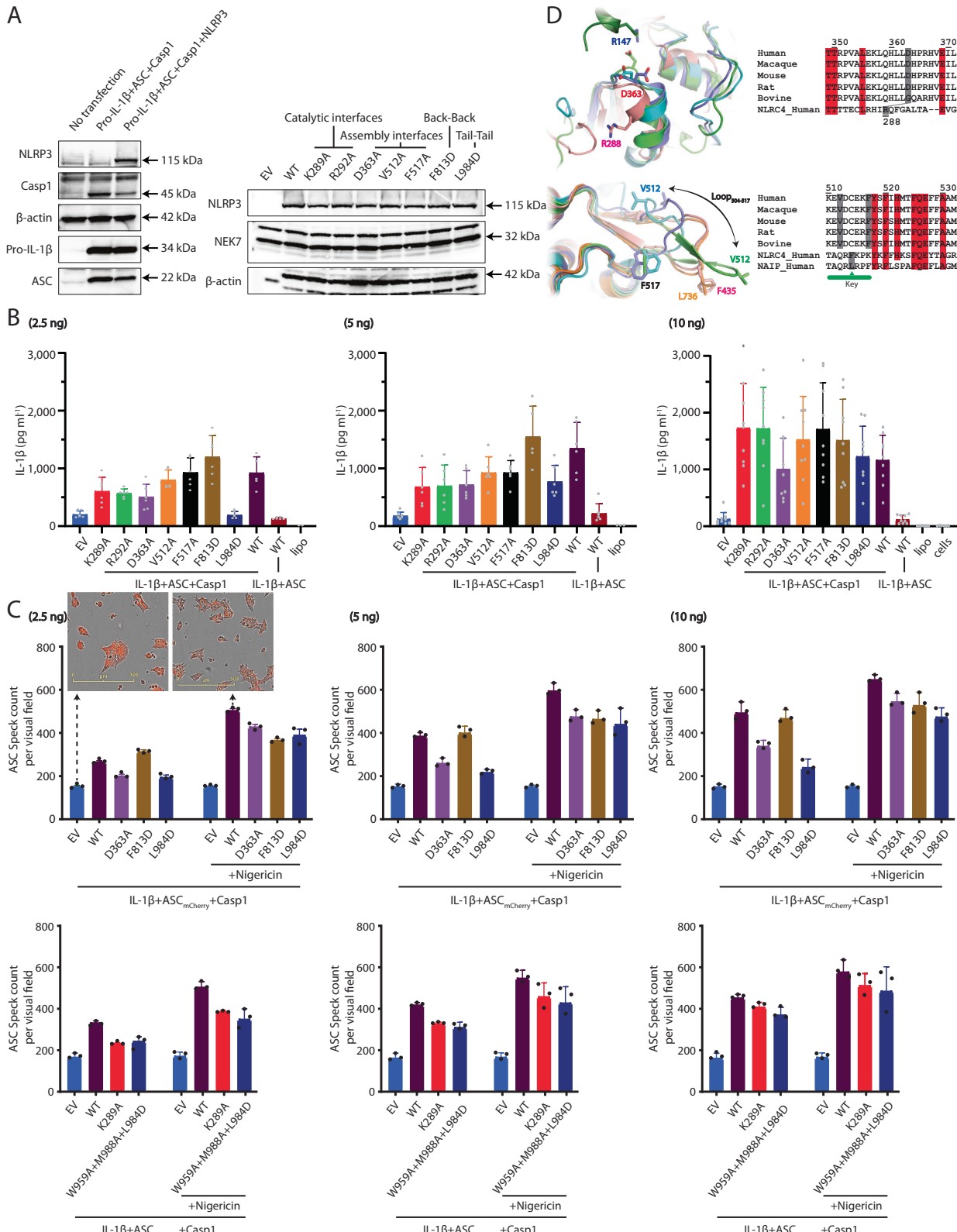

carrying these mutations, as well as the wild-type (WT) NLRP3, were transfected into HEK293T cells, respectively (Fig. 4B). Compared to WT NLRP3, the expression levels of NLRP3 and NEK7 were not affected by these substitutions as seen by Western blot analysis (Supplementary Fig. 8B). All substitutions, except for Phe$_{813}$Asp, exhibited a reduced induction of IL-1β signaling upon transfection with 2.5 ng of vectors into the cells. However, these differences comparing to the WT

were rectified when higher quantities of vectors were transfected, suggesting that the oligomerization state induced by the increased NLRP3 expression could potentially overcome the substitution effects (Fig. 4B).

The Leu$_{984}$Asp substitution, which was intended to disrupt the Tail-Tail interactions in the octamer, displayed a compromised induction of IL-1β upon transfection with 2.5 or 5 ng of vectors into the

**Fig. 4 | Interfaces in the open octamer are critical in the NLRP3 activation and assembly. A** Mutations at the interfaces did not impair NLRP3 expression level. The component proteins were detected using Western blots. These experiments were repeated twice. **B** Cellular assay showing induction of IL-1β signaling upon transfection with 2.5, 5, or 10 ng of WT or mutant FL NLRP3 vectors. EV, empty vector control. Bars indicate the mean, and the data points from six independent experiments are shown. **C** Structural and sequence alignments around the catalytic surface around $Asp_{363}$ (top) and the $Loop_{504-517}$ region (lower), respectively. Closed (PDB ID: 7LFH), open (this study), activated (PDB ID: 8EJ4) NLRP3, NLRC4 (PDB ID:

8FW2), and NAIP (PDB ID: 8FVU) were shown as cartoon and in cyan, slate, light green, pink, and orange, respectively. The adjacent activated NLRP3 protomer was colored in dark green. Side chains of selected residues were shown in sticks in the structural alignments and highlighted with filled gray boxes in the sequence alignments. **D** Fluorescent mCherry measurements in HEK293TASC-mCherry cells transfected with empty vector (EV), NLRP3 WT, or mutants, with/without Nigericin treatment. Two cell images with scale bars were shown as reference. Bars indicate the mean, and the data points from three independent experiments are shown. Source data are provided as a Source Data file.

cells (Figs. 2D, 4B). Triple substitutions ($Trp_{959}Ala+Leu_{984}Asp+Met_{988}Ala$) at the Tail-Tail interface further decreased IL-1β induction with 2.5 ng of vectors (Supplementary Fig. 8C). In addition, $Lys_{289}$ and $Arg_{292}$ are at the Head-Face interface in the octamer and their Ala substitutions at these sites displayed pronounced effects at 2.5, or 5 ng of vector transfection, with subtle effects at 10 ng transfection (Figs. 2C, 4B). $Leu_{984}$, $Lys_{289}$, or $Arg_{292}$ do not form direct contacts with the NEK7 or neighboring protomers in the NLRP3 inflammasome. $Leu_{984}$, $Lys_{289}$ are conserved residues crossing the NLRP3 family (Supplementary Fig. 9). These mutational findings indicate that the Tail-Tail and Head-Face interfaces within the open octamer play critical roles in the NLRP3 activation.

$Lys_{289}$, $Arg_{292}$, $Asp_{363}$, and $Val_{512}$, $Phe_{517}$ are situated on opposite sides of NACHT in the closed NLRP3 structures but align on the same surface in the open and activated NLRP3 structures (Supplementary Fig. 10A, B). A structural overlay of the NBD domains from NLRP3 and open NLRC4 structures showed the close residue to $NLRP3$-$Asp_{363}$ is $NLRC4$-$Arg_{288}$ (Fig. 4C top panel, Supplementary Fig. 9). $Arg_{288}$ forms a critical salt bridge with $Asp_{125}$ from a following protomer in the NLRC4 inflammasome structure. A $NLRC4$-$Arg_{288}Ala$ substitution resulted in stunted inflammasome formation[27]. Similarly, $NLRP3$- $Asp_{363}Ala$ substitution perturbed the salt-bridge formation with $Arg_{147}$ from a neighboring protomer in the activated NLRP3 inflammasome, resulting in the reduced IL-1β signaling inductions (Fig. 4C, Supplementary Fig. 10C). $Val_{512}$, and $Phe_{517}$ are located in a loop region (a.a. 504-517), which undergoes significant conformational changes (Fig. 4C lower panel, Supplementary Figs. 4, 10C). In the closed cage and open octamer structures, $Loop_{504-517}$ curls back to its own NACHT domain. In the activated NLRP3 inflammasome structure, $Loop_{504-517}$ extends and forms hydrophobic interactions with the following NLRP3 protomer through $Val_{512}$ (Supplementary Fig. 10C). A $Val_{512}Ala$ substitution may weaken the inflammasome assembly by reducing its hydrophobicity. Consistently, the corresponding residue of $Val_{512}$ in the mouse NLRC4 is $Leu_{435}$ ($Phe_{435}$ in human NLRC4) (Fig. 4C, Supplementary Fig. 9). Mouse $NLRC4$-$Leu_{435}Asp$ mutation still displayed flagellin-induced interaction with NAIP5 but failed to form higher-order oligomeric complex[27]. $Phe_{508}$ works together with $Phe_{517}$ anchoring $Loop_{504-517}$ and assists inflammasome assembly (Supplementary Fig. 10C). A $Phe_{508}Ala$ substitution may disrupt the $Loop_{504-517}$, thus affecting the inflammasome assembly. $Asp_{363}$, $Val_{512}$, and $Phe_{517}$ did not directly participate in the oligomer formation in the open octamer, indicating that substitutions in these residues affect the NLRP3 inflammasome assembly rather than activation.

Remarkably, $Phe_{813}$ is positioned at the Back-Back interface and its substitution to an Aspartic acid resulted in a subtle increase in IL-1β induction upon vector transfection with 2.5, 5, or 10 ng, contrary to the trend observed with other substitutions (Figs. 2B, 4B). Additionally, further substitutions at the Back-Back interface (Supplementary Fig. 8C) progressively decreased IL-1β induction with 2.5 ng vector transfection. These findings suggest that the Back-Back interaction behaves differently compared to substitutions at other interfaces within the open octamer (see discussion).

## Oligomerization assists NLRP3 ASC speck formation, bypassed by Nigericin

We further explored the influence of oligomerization states on NLRP3 ASC speck formation[28,29]. The mCherry-tagged ASC was stably transduced into HEK293T cells, which were then transiently transfected with different amounts (2.5 ng, 5 ng, and 10 ng) of human NLRP3 constructs carrying mutations in the oligomerization interfaces (Fig. 4D). Evaluation of mCherry signals was performed using the Incucyte S3 (2022B Rev2 software) with basic analyzer and surface-fit analysis. Without NLRP3, mCherry signals diffused within the cells. In contrast, in the presence of NLRP3 with or without Nigericin, we observed intense mCherry clusters, signaling ASC speck formation. Consistent with our IL-1β signaling experiments after NLRP3 transfection, mutations in the oligomerization interfaces influenced ASC speck formations, while, the Back-Back interface mutation (F813D) enhanced ASC speck formation. Notably, Nigericin significantly enhanced ASC speck formation in both WT and mutated NLRP3 (Fig. 4D), along with heightened IL-1β signaling (Supplementary Fig. 8C). Our cellular results collectively highlight the critical role of oligomerization in NLRP3 activation and inflammasome assembly. Furthermore, Nigericin was observed to potentially bypass the oligomerization requirement, efficiently activating NLRP3 and inducing ASC speck formation.

## NEK7 dissociates NLRP3 oligomers into NEK7/NLRP3 dimers

We next introduced the recombinant purified NEK7 into the $NLRP3_{\Delta PYD/+ATP}$ in presence or absence of MCC950. NEK7 formed a complex with $NLRP3_{\Delta PYD/+ATP/+/-MCC950}$ as confirmed by SEC and SDS-PAGE (Supplementary Fig. 1A). Cryo-EM analysis showed non-productive aggregations without MCC950. With MCC950, $NLRP3_{\Delta PYD/+ATP/+MCC950}$ exhibited two distinct stable populations within one cryo-EM dataset, which were further subjected to refinement and reconstruction individually by applying D3 or C2 symmetry, respectively (Fig. 5A, B, Supplementary Figs. 2, 3, Supplementary Table 1). The D3 hexamer is identical to the previously identified $NLRP3_{\Delta PYD/+ADP/+MCC950}$ complex structure (PDB ID: 7VTP, 7ZGU), while the C2 structure is composed of a $NLRP3_{\Delta PYD/+ATP/+MCC950}$ dimer via the back-back interaction with NEK7 densities on each concave side of LRRs (Fig. 5B, Supplementary Figs. 2, 3C). ATP and MCC950 densities were observed in the NACHT domains of closed $NLRP3_{\Delta PYD}$ hexamer and NEK7/$NLRP3_{\Delta PYD}$ dimer (Supplementary Fig. 3D). NEK7 engagement did not impact the binding poses of MCC950 or ATP/ADP in the NACHT domain, nor alter the closed conformation of $NLRP3_{\Delta PYD/+ATP/+MCC950}$ (Fig. 5C, Supplementary Fig. 11)[3]. A structural overlay using one of the $NLRP3_{\Delta PYD/+ATP/+MCC950}$ protomers revealed that the binding of NEK7 slightly opens the NACHT domain, meanwhile altering the conformation of the $NACHT$-$Loop_{617-628}$ (a.a. 617−628) (Fig. 5C, Supplementary Fig. 11).

The calculated NEK7 binding interface with the closed $NLRP3_{\Delta PYD/+ATP/+MCC950}$ measures -2249.2 Å[2], involving around 74 contact residues (Supplementary Fig. 6B, C, Supplementary Table 3). In the closed $NLRP3_{\Delta PYD/+ATP/+MCC950}$ hexamer, the NEK7 binding site at LRR is open accessible. A structural overlay using the protomers from the closed $NLRP3_{\Delta PYD/+ATP/+MCC950}$ hexamer and NEK7/$NLRP3_{\Delta PYD/+ATP/+MCC950}$ dimer reveals that the N-lobe of NEK7 clashes with the adjacent NLRP3 protomer within the closed

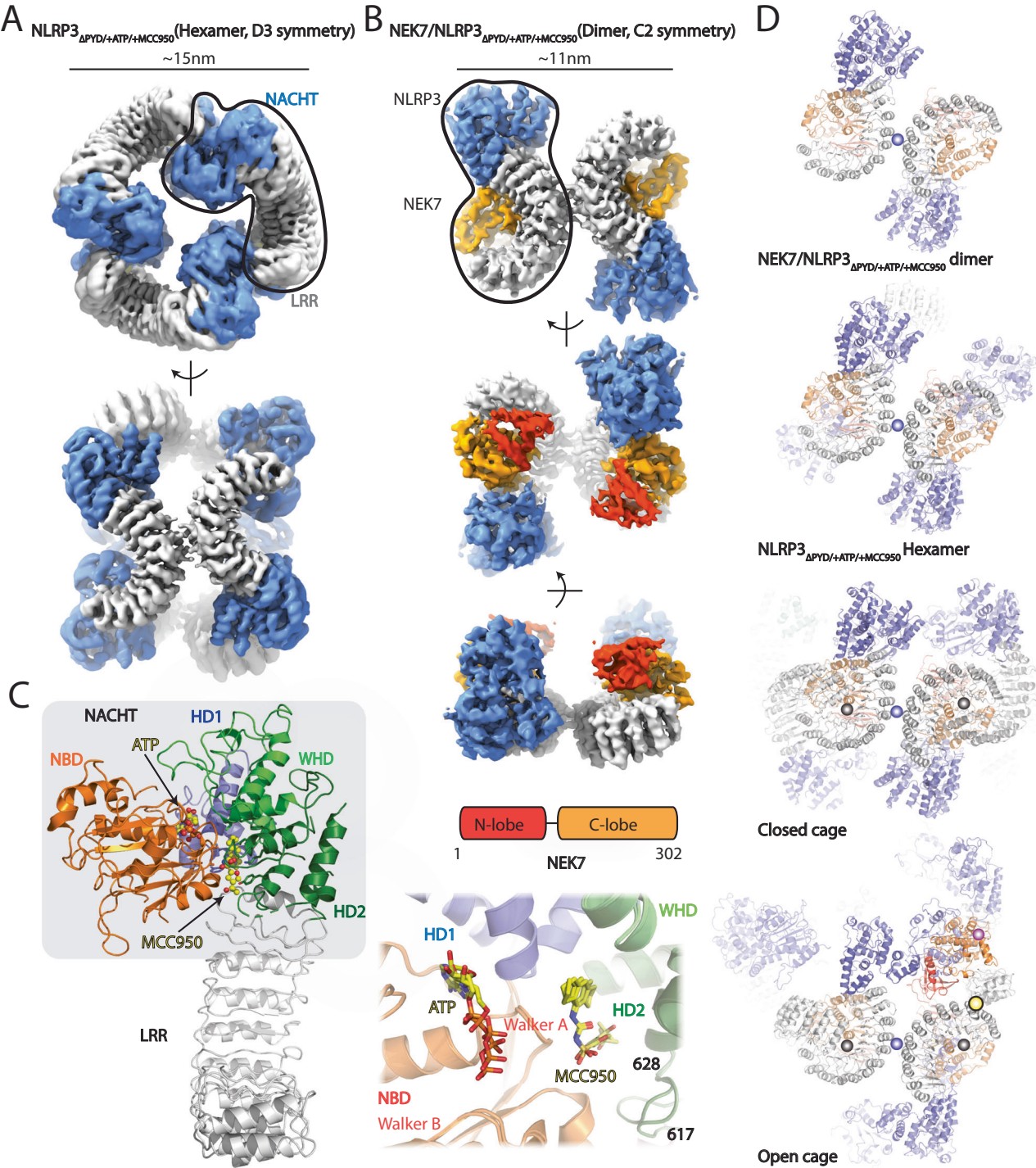

**Fig. 5 | Cryo-EM structures of NLRP3$_{\Delta PYD/+ATP/+MCC950}$ in complex with NEK7. A, B** Cryo-EM maps of closed NLRP3$_{\Delta PYD/+ATP/+MCC950}$ hexamer and NEK7/NLRP3$_{\Delta PYD/+ATP/+MCC950}$ dimer, respectively. Domain organization of NEK7 was shown. The NACHT, LRR, NEK7-N-lobe, and NEK7-C-lobe were colored in slate, gray, red, and orange, respectively. One protomer was highlighted, respectively. **C** Structure of closed NLRP3$_{\Delta PYD/+ATP/+MCC950}$ protomer within the hexamer (Left) and structural overlay of closed NLRP3$_{\Delta PYD/+ATP/+MCC950}$ hexamer and NEK7/NLRP3$_{\Delta PYD/+ATP/+MCC950}$ dimer at MCC950 binding site (Right). ATP and MCC950 were shown as sticks. NACHT subdomains NBD, HD1, WHD, and HD2 were colored in orange, slate, light green, and dark green, respectively. Loop$_{617-628}$ was highlighted. **D** From top to down, structure of NEK7/NLRP3$_{\Delta PYD/+ATP/+MCC950}$ dimer, docking of NEK7 to the concave sides of LRRs in the closed hexamer (this study), FL cage (PDB ID: 7LFH) and open octamer (this study). The Back-Back, Face-Face, Head-Face, and Tail-Tail interactions were highlighted using magenta, gray, pink, and yellow spheres, respectively. The NACHT, LRR, and NEK7 were colored in slate, gray, and orange, respectively.

NLRP3$_{\Delta PYD/+ATP/+MCC950}$ hexamer, leading to the closed hexamer dissociation. In the closed FL NLRP3 cages, the NEK7 binding interface is shielded by the Face-Face interaction (Supplementary Fig. 6B–D). However, both the calculated Face-Face interface and the contact residues are smaller than those of the NEK7/NLRP3 binding interface

(Supplementary Table 3). In the NLRP3$_{\Delta PYD/+ATP/-MCC950}$ open octamer structure, the NEK7 binding surface can be occupied by either the neighboring protomer's LRR resulting in a Face-Face interaction or the neighboring protomer's NACHT forming a Head-Face interaction (Figs. 2C, E, 3A, D). Both the calculated area and contact residues at the

Face-Face and Head-Face interfaces are approximately half the values of the NEK7/NLRP3 binding interface (Supplementary Fig. 6B, C, Supplementary Table 3). Docking NEK7 at the LRR binding sites in NLRP3 closed cages or open octamer results in clashes with the neighboring protomer (Fig. 5D, Supplementary Fig. 6D). These findings suggest that the presence of NEK7 leads to the dissociation of large NLRP3 oligomers, both closed and open, into NEK7/NLRP3 dimers, while the Back-Back interaction is maintained.

## Discussion

Based on available cryo-EM structures, we propose a activation mechanism for NLRP3 (Fig. 6). This involves a two-step process starting with priming, which triggers the NLRP3 transcriptional upregulation and oligomerization forming the closed cages. The formation of the closed NLRP3 cages was driven by LRR-mediated repeated Face-Face and Back-Back interactions, while NACHTs do not directly interact (Fig. 3C)[5,6]. Structural comparison between human and mouse closed cages shows Face-Face interface plasticity and relatively stable Back-Back interactions (Fig. 3A, B, Supplementary Fig. 6A, B). The closed NLRP3 cages can be activated by a second signal, such as ATP (Supplementary Fig. 1C). From the closed NLRP3 structures, ATP forms interactions with NBD, HD1, and WHD (Fig. 5C). ATP hydrolysis potentially induces NACHT perturbations enabling conformational changes. Additionally, the open octamer provides an LRR scaffold with a large negatively charged surface, attracting the positively charged NACHT-NBD interface, facilitating NACHT conformational changes (Fig. 3D, Supplementary Fig. 5). The NACHT opening and cage rearrangement from closed to open may occur simultaneously or mutually support one another (Supplementary Movie 2, Fig. 3D). Note that the PYD is not involved in this proposed NLRP3 activation mechanism. In the closed FL NLRP3 cages, PYDs were either found at the center of cages or exhibited dynamic movement around the cages[5,6]. In the open octamer, MD simulation results indicate that all PYDs surround the octamer, following the N-terminal trend of NBD (Supplementary Fig. 7A, Supplementary Movie 3). This suggests that PYD does not impact octamer formation; instead, it may be located at a binding site on NBD, which is only present in the open and activated forms (Supplementary Fig. 7, Supplementary Movie 3).

The open octamer features Face-Face interactions among receptor protomers (Figs. 3A, 2(2′), and 3(3′)) and Tail-Tail interactions among catalytic protomers (Figs. 3A, 1(4), and 1′(4′)). Receptor protomers interact Back-Back with catalytic protomers and donate their NBD to form Head-Face interactions with the other catalytic protomer (Figs. 2A, 3D). Two novel interfaces (Head-Face and Tail-Tail) observed in the open octamer play critical roles in NLRP3 activation and inflammasome assembly (Fig. 2C, D). Point mutations that disrupt the octamer oligomerization compromise IL-1β response, but this inhibitory effect can be overcome by increasing NLRP3 expression level, demonstrating the importance of this oligomerization in NLRP3 activation and inflammasome formation (Fig. 4B).

MCC950 inhibits ATP hydrolysis and acts as molecular glue, interacting with NACHT's subdomains to lock it in the closed form (Figs. 5C, 6, Supplementary Fig. 1C). The closed and open NLRP3 structures underwent Target MD simulations to capture conformational transitions during NLRP3 activation (Supplementary Fig. 12A,B, Supplementary Movie 4). The MD results indicated that Nucleotide/MCC950 binding sites were impacted first, followed by NBD-WHD-HD2 separation with HD1 as the pivot point. The initial NBD-WHD-HD2 separation likely incurred high energy cost due to the dismissed interaction interfaces between NBD and WHD-HD2. Multiple GOF mutations around nucleotide/MCC950 binding sites at the interfaces can lower the energy barrier during activation, leading to NLRP3 constitutive activation (Supplementary Figs. 9, and 12C)[30,31].

The acidic-loop_{689-698} was found to shield the LRR-catalytic surface in the closed human cage (PDB ID: 7PZC) and was not resolved in the open octamer[5] (Supplementary Fig. 13A). The acidic loop can hinder receptor-NBD binding to catalytic-LRR concave surface (Supplementary Fig. 6C). It can also impede NEK7 binding to the LRR. Indeed, deletion of the acidic loop showed a marked loss of auto-inhibition, facilitating NLRP3 activation[5]. However, we cannot eliminate the possibility that altering properties or position of the acidic loop might aid NACHT opening. Two GOF mutations in this acidic-loop region were linked to NLRP3 constitutive activation (Supplementary Fig. 9)[20]. NEK7 engagement or Face-Face, and Head-Face interactions that were observed in the open octamer may help reposition the acidic loop to aid NACHT opening (Supplementary Fig. 13A). Indeed, the binding of NEK7 altering the acidic loop slightly opened the NACHT domain even in the presence of MCC950 (Supplementary Figs. 11, 13A).

The open NLRP3 octamer may form during the *trans*-Golgi network dispersion or transporting to the centrosome, where it can engage with centrosomal NEK7 to form the active NLRP3 inflammasome speck[32–35]. NEK7 does not participate in the proposed NLRP3 activation process but may help to dissociate the octamer into open NEK7/NLRP3 dimers for subsequent inflammasome assembly

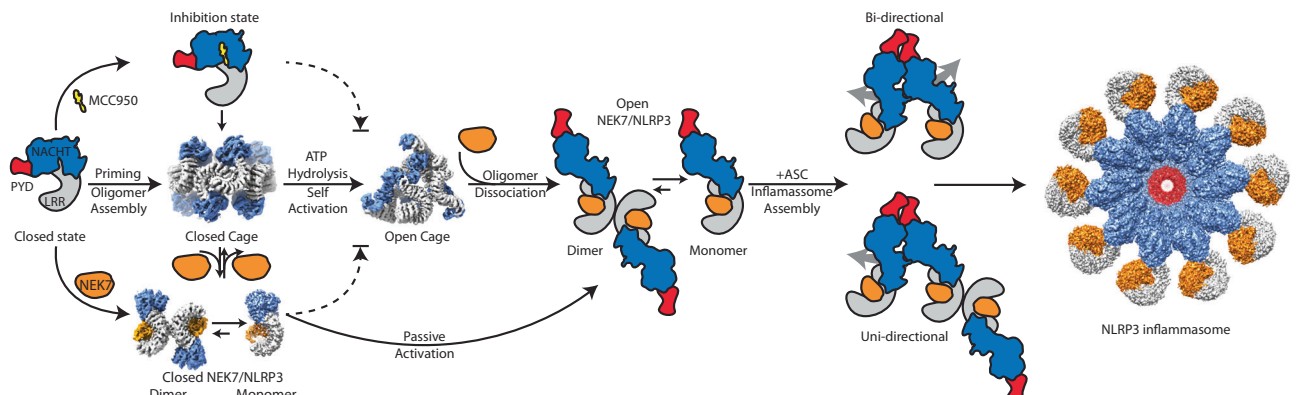

**Fig. 6 | Proposed NLRP3 activation and uni- or bi- directional inflammasome assembly mechanisms.** The priming step increases NLRP3 levels, which leads to NLRP3 oligomerization. The activation of NLRP3 occurs through ATP hydrolysis and oligomer reorganization, forming an open octamer. These closed or open oligomerization states are favorable for trafficking on microtubules to the MTOC. NEK7 then dissociates the oligomer into NEK7/NLRP3 binary complexes in equilibrium with dimer and monomer. The open NEK7/NLRP3 allows for uni- or bi-directional inflammasome assembly as indicated by the gray arrows, while the closed NEK7/NLRP3 can also be passively activated by the open NEK7/NLRP3. MCC950 inhibits NLRP3's ATPase hydrolysis activity, binds to all NACHT subdomains, and locks it in a closed conformation. The closed cage, NEK7/NLRP3 monomer, and the fully activated NLRP3 inflammasome were derived from PDB ID: 7LFH, 6NPY, and 8EJ4, respectively. The dashed arrows illustrate how MCC950 or NEK7's early dissociation of the closed cage prevents NLRP3 activation.

(Figs. 5D, 6, Supplementary Fig. 6B–D, 13B, Supplementary Table 3). NEK7 binds to the concave side of LRR, which largely overlaps with the (Head)Face-Face interactions in the open octamer or Face-Face interactions in the closed cages (Fig. 5D, Supplementary Fig. 6D). Based on interface area and residue count (Supplementary Table 3), NEK7 binding to the LRR site can probably dissociate both closed cages and open octamers. Hence, the presence of NEK7 should be tightly regulated to interact with the open octamer, to avoid dissociation of the closed cage, resulting in non-productive closed NEK7/NLRP3 (Fig. 6, Supplementary Fig. 13B).

The Back-Back interaction has a dual effect on the NLRP3 activation and inflammasome assembly. It is observed in the closed and open NLRP3 oligomers but absent in the activated NLRP3 inflammasome (Figs. 2A, B, 3). They are necessary for NLRP3 oligomer formation but the observed NEK7/NLRP3 dimer, facilitated by Back-Back interaction, can also impede NLRP3 inflammasome assembly by covering the receptor surface for subsequent NEK7/NLRP3 recruitment (Fig. 5B, Supplementary Fig. 13B). The NEK7/NLRP3 most likely exists in a dynamic equilibrium between monomers[3] (PDB ID: 6NPY) and dimers (this study) (Fig. 6, Supplementary Fig. 13B), which allows the closed cages with even-numbered protomers (6-, 8-, 10-, 12-, 14-, or 16-mers)[4–6] to eventually form the disk-shaped inflammasomes with even or odd- numbered protomers (10 or 11- mers NLRP3 inflammasome complex)[23] (Fig. 6, Supplementary Fig. 13B). A point mutation (Phe$_{813}$Asp) disrupting the Back-Back interaction shifts the balance towards the open NEK7/NLRP3 monomers and enhances IL-1β signaling in a cell model (Fig. 4B). Additional substitutions at the Back-Back interface (Supplementary Fig. 8C) progressively decreased IL-1β levels, suggesting they impact both closed and open cage formation and oligomerization-facilitated NLRP3 activation.

Compared to the closed form, the open form NLRP3 exposes a large catalytic surface for inflammasome assembly (Supplementary Figs. 10A,B, 12A), but an unsupervised mixture of the open NEK7/NLRP3 may cause aggregation. Unlike NLRC4, a disk-shaped NLRP3 structure was reported from the co-expression of FL NLRP3 with NEK7, followed by ASC PYD incubation[23]. The PYD domains and ASC can form helical conformations, respectively, and guide the inflammasome assembly. Here, we captured an intermediate state of NLRP3 activation, which can occur in the absence of PYD, NEK7, or ASC (Fig. 6, Supplementary Fig. 12B). Our cellular studies were based on transiently transfecting NLRP3 mutants in HEK293T cells. Yet, we can't rule out alternate pathways in NLRP3 inflammasome activation, like NLRC4 crosstalk, Nigericin, or lipopolysaccharide (LPS) activation in human macrophages[21]. Our cellular assay demonstrated that Nigericin more effectively induced NLRP3 activation and subsequent IL-1β signaling (Fig. 4D, Supplementary Fig. 8C), implying that Nigericin may promote NLRP3 activation independently of oligomeric states. The authors hope that our present studies will serve as inspiration for future, advanced studies in primary human cells.

Unlike the uni-directional assembly of NLRC4 inflammasome, which is initiated from the activated NAIP, the NLRP3 inflammasome can be assembled bi-directionally from the open NEK7/NLRP3 (Fig. 6, Supplementary Fig. 13B). Furthermore, the closed NEK7/NLRP3 monomers can be passively activated through the exposed large catalytic surface (Supplementary Figs. 10A, B, 12A) of the open NEK7/NLRP3 binary complex initiating the inflammasome assembly similar to NLRC4 (Fig. 6). The activated NEK7/NLRP3 protomers can assemble into fragments, which later combine to form a disk-shaped inflammasome.

## Methods
### Recombinant proteins
All purifications were done at 4 °C. All gene synthesis and cloning were done at Epoch Life Science.

DNA encoding human NEK7 (UniProt Q8TDX7) engineered with kinase mutations K63M/K64M and an N-terminal 6His.SUMO tag was inserted into a pET28a vector (Novagen). 6His.SUMO.NEK7$_{K63M/K64M}$ was overexpressed in BL21(DE3) *E. coli* with 0.1 mM IPTG induction for 19 h at 16 °C. Cell paste was incubated with lysozyme (Novagen), and benzonase (Millipore), and 6His.SUMO.NEK7$_{K63M/K64M}$ was released from the cells by pressure drop lysis in 25 mM HEPES pH = 7.5 0.3 M NaCl 20 mM imidazole 5% glycerol 0.1 mM TCEP with Complete EDTA protease inhibitors (Sigma Aldrich). 6His.SUMO.NEK7$_{K63M/K64M}$ was then captured from clarified lysate supernatant with HisPur Ni resin (Thermo Fisher Scientific). The imidazole eluate was treated with Ulp1 SUMO hydrolase, desalted into 25 mM HEPES pH = 7.5 0.3 M NaCl 0.1 mM TCEP 5% glycerol and then flowed through HisTrap HP (Cytiva) to remove the 6His.SUMO tag. NEK7$_{K63M/K64M}$ was concentrated and further purified by Superdex 200.

DNA encoding human NLRP3 (UniProt Q96P20) was engineered to make two delta PYD constructs (NLRP3$_{ΔPYD}$); FLA-G.6His.MBP.TEV.CfaC.NLRP3_130.K131F.M132N-1036 and MBP.TEV.NLRP3_130_1036. These constructs were cloned into a PVL1393 transfer vector (Expression Systems) and cotransfected into Sf9 insect cells with Best Bac 2.0 (Expression Systems, Catalog number 94-001 F). The recombinant baculovirus generated was used to infect Sf9 insect cells for 56 to 96 h at 27 °C. The NLRP3$_{ΔPYD}$ constructs were released from cells by sonication in 25 mM HEPES pH = 7.4 0.4 M NaCl 0.4% CHAPS 20% glycerol 0.2 mM TCEP with Complete EDTA protease inhibitors (Roche) and then captured on to amylose resin (New England BioLabs Inc.) from the clarified lysate supernatant. The amylose resins were washed with 2 M NaCl followed by a 10 mM ATP-MgCl$_2$ and eluted with 20 mM maltose. MBP.TEV.NLRP3_130_1036 was directly processed on Superose 6 in 25 mM HEPES pH = 7.4 0.4 M NaCl 0.2% CHAPS 0.2 mM TCEP 20% glycerol and delivered for ATPase and nanoDSF assays. The FLAG.6His.MBP.TEV.CfaC.NLRP3_130.K131F.M132N_1036 amylose eluates, while undergoing tag removal with CfaN peptide[36], were treated either +/−10 fold molar excess MCC950 (CAS Number: 256373-96-3) and +/−2 fold molar excess NEK7$_{K63M/K64M}$. NLRP3$_{ΔPYD/+ATP/(+/-)MCC950}$ was processed on Superose 6 in 25 mM HEPES pH = 7.4 0.4 M NaCl 0.2% CHAPS 0.2 mM TCEP; retained peak fractions were delivered for CryoEM. NLRP3$_{ΔPYD/+ATP/+NEK7/(+/−)MCC950}$, was processed on Superdex 200 in 25 mM HEPES pH = 7.4 0.4 M NaCl 0.2% CHAPS 0.2 mM TCEP; retained peak fractions were delivered for Cryo-EM.

### NanoDSF
For Nanoscale Differential Scanning Fluorometry (NanoDSF) measurements, a Prometheus NT.48 (NanoTemper Technologies GmbH, Munich, Germany) was used to analyze protein thermal stability in response to ligand binding. Purified recombinant human NLRP3$_{ΔPYD}$ (1 μM) was incubated at room temperature with and without 100 μM of MCC950 + 1 mM ATP (Sigma A2383) in 12 μL reaction volume for 30 min in reaction buffer containing 25 mM HEPES pH 7.4, 400 mM NaCl, 0.2% CHAPS, 0.2 mM TCEP. Assays were prepared in Greiner 384 well non-binding black plates (Greiner 784900) and transferred to the instrument using the Prometheus NT.Plex nanoDSF Grade Standard Capillary Chips (PR-AC002). The instrument excites samples at 280 nm and records intrinsic protein fluorescence at 330 and 350 nm, the lambda maxes associated with buried and exposed tryptophan residues, respectively. Measurements were taken over a 20−90 °C thermal gradient with a 1 °C per minute ramp rate. The 350 nm/330 nm fluorescence ratio was plotted versus temperature and the first derivative of this plot was used to determine the melting temperature (T$_M$) under each condition. Data analysis was done by PR.ThermControl, version 2.1.6 by NanoTemper.

### NLRP3 ATPase activity and inhibition
For ATPase activity assay, purified recombinant human NLRP3$_{ΔPYD}$ (0.5 μM) was incubated at room temperature with/without indicated concentration of MCC950 for 30 min in the reaction buffer containing 25 mM HEPES pH 7.4, 0.2 mM NaCl, 5 mM MgCl$_2$, 0.01% CHAPS, 0.1 mM

TCEP. ATP (100 μM, Ultra-Pure ATP) was then added, and the mixture was further incubated at 37 °C for another 5 h. The amount of ATP converted into adenosine diphosphate (ADP) was determined by luminescent ADP detection with an ADP-Glo Kinase Assay kit (Promega, Madison, MI, USA) according to the manufacturer's protocol. The results were expressed as percentages of residual enzyme activity to the vehicle-treated enzyme.

## Grid preparation and data acquisition

3.5 μL of purified 3-4 mg/ml NLRP3$_{\Delta PYD/+ATP/-MCC950}$ or NLRP3$_{\Delta PYD/+ATP/+MCC950}$ with/without NEK7 complex was applied to the plasma-cleaned (Gatan Solarus) Quantifoil 1.2/1.3 holey gold grid and subsequently vitrified using a Vitrobot Mark IV (FEI Company). Cryo grids were loaded into a Titan Krios transmission electron microscope (Thermo Fisher Scientific) with a post-column Gatan Image Filter (GIF) operating in nanoprobe at 300 keV with a Gatan K3 Summit direct electron detector and an energy filter slit width of 20 eV. Images were recorded with Leginon in counting mode. All details corresponding to individual datasets are summarized in Supplementary Table 1.

## Electron microscopy data processing

Dose-fractioned movies were gain-corrected, and beam-induced motion correction using MotionCor2[37] with the dose-weighting option. The particles were automatically picked from the dose-weighted, motion-corrected average images using Relion 3.0[38]. CTF parameters were determined by Gctf [39]. Particles were then extracted using Relion 3.0 and followed by several rounds of 2D and 3D classifications to select the homogenous particles. One stable class of particles was selected from individual NLRP3$_{\Delta PYD/+ATP/+MCC950}$ or NLRP3$_{\Delta PYD/+ATP/-MCC950}$ datasets and followed by Relion's 3D auto-refine with D3 or D2 symmetry imposed, respectively. From the dataset of the NLRP3$_{\Delta PYD/+ATP/+MCC950}$ complex with NEK7, two stable populations of particles with distinctive features were further subjected to refinement and reconstruction individually by imposing D3 or C2 symmetry, respectively The final particle stacks with the parameters from the last iteration of the 3D auto-refine were inputted into CryoSPARC[40]. Particles were then subjected to symmetry expansion, and then one protomer was masked and subjected to local refinement with C1 symmetry with the standard deviation of prior over rotation/shift parameters set to 3 ° and 2 Å, respectively. 3D classifications and 3D refinements were started from a 60 Å low-pass filtered version of an ab initio map generated with Relion 3.0. All resolutions were estimated by applying a soft mask around the protein complex density and based on the gold-standard (two halves of data refined independently) FSC = 0.143 criterion. Prior to visualization, all density maps were sharpened by applying different negative temperature factors using automated procedures, along with the half maps, were used for model building. Local resolution was determined using ResMap[41] (Supplementary Figs. 2, 3). The number of particles in each data set and other details related to data processing are summarized in Supplementary Fig. 2, and Supplementary Table 1.

## Model building and refinement

The EM density map of the open octamer reveals distinct features corresponding to the LRR, while the closed form NLRP3 structures only allows for LRR docking, with the NACHT domain extending beyond the map. Furthermore, during the structural determination and modeling of the NLRP3 open octamer, no activated form structure was available as an initial model. An Alphafold2 (AF2) FL NLRP3 model (AF-Q96P20-F1) was used as an initial model. Individual subdomain structures (NBD, HD1, WHD, HD2, and LRR) (Supplementary Fig. 4) were extracted from the AF2 model and then docked onto one protomer of the open octamer EM density map using Chimera[42]. These docked subdomains were merged into a single protomer, removing

the disordering loops without the EM density support, and followed by manual adjustment using COOT[43]. The complete NLRP3 octamer structure was created from a single protomer and then expanded into a symmetric structure (D2) using Chimera software. The initial templates of the closed NLRP3 and NEK7 were derived from homology-based models calculated by SWISS-MODEL[44]. The models were docked into the EM density maps (closed NLRP3$_{\Delta PYD/+ATP/+MCC950}$ hexamer, and NEK7/NLRP3$_{\Delta PYD/+ATP/+MCC950}$ dimer) using Chimera[42] and followed by manual adjustment using COOT[43].

Each model was independently subjected to global refinement and minimization in real space using the module *phenix.real_space_refine* in PHENIX[45] against separate EM half-maps with default parameters. The model was refined into a working half-map, and improvement of the model was monitored using the free half-map. The geometry parameters of the final models were validated in COOT and using MolProbity[46] and EMRinger[47]. These refinements were performed iteratively until no further improvements were observed. The final refinement statistics are provided in Supplementary Table 1. Model overfitting was evaluated through its refinement against one cryo-EM half map. FSC curves were calculated between the resulting model and the working half map as well as between the resulting model and the free half and full maps for cross-validation. Figures were produced using PyMOL[48] and Chimera.

## Mutation assay

Expression vectors (pRD) were designed and ordered replacing indicated amino acids by alanine or aspartic acid. Hek293T cells were cultured using DMEM supplemented with 10% heat-inactivated fetal bovine serum, 1% penicillin-streptavidin and 1% L-glutamine and plated in 96 well plate at 40.000 cells per well to allow attachment overnight. The next day, fresh medium was added, and transfection was performed using Lipofectamine 2000 (Thermo Fisher Scientific)[49] according to the manufacturer's protocol with 200 ng plasmid per well (2.5, 5, or 10 ng of NLRP3, 0.3 ng ASC, 50 ng pro-IL-1β, and 1 ng Casp1). After 24 hrs, the supernatant was collected and IL-1β (Thermo Fisher Scientific) was detected via Luminex platform. For western blotting, overexpression was performed in 12 well plates and lysates used to detect NLRP3 (Cryo-2, 045088, US Biologicals, L21042362), ASC (pAL177, Adipogen, A4272210), Caspase-1 (Bally-1, US Biological, 44634), IL-1β (GTX4034, Genetex, 43110), NEK7 (133514, Abcam, GR3419385-10) and β-actin (sc-47778-HRP, Santa Cruz Biotechnology, J2020). sc-2004 (Santa Cruz Biotechnology, INC.) targets goat anti-rabbit IgG-HRP; 31430 (Termofisher Scientific) targets Goat anti-Mouse IgG (H + L) Secondary Antibody, HRP. Primary antibodies (Cryo-2, pAL177, GTX4034, 133514, and sc-47778-HRP) were used at a 1:1000 dilution, except for Bally-1, which was used at 1:500. Secondary antibodies (sc-2004, and 31430) were used at a 1:5000 dilution.

## Global Molecular Dynamics (MD) simulation

To assess the impact of PYD domains on the open NLRP3 octamer, we conducted molecular dynamic (MD) simulations using a chimera model derived from the open NLRP3$_{\Delta PYD/+ATP/-MCC950}$ structure and the PYD subdomain model from the AF2 FL NLRP3 model (AF-Q96P20-F1). The PYD$_{1-147}$ model was isolated from the AF2 model and fused to each individual protomer within the open octamer using the α-helix$_{136-147}$ as the reference. The resulting chimera FL NLRP3 open octamer was subjected to the default relaxation protocol in Maestro and followed by a 100 ns MD simulation. The simulation was run at 300 K and 1 bar for 100 ns using the Desmond simulation package[50] in Schrodinger 2021-1 (www.schrodinger.com). The systems were protonated at neutral pH and centered in a cubic box such that the minimum distance from any protein atom to the box wall was 10 Å. The box was solvated using SPC[51] water molecules and counter ions were added to neutralize

the system. OPLS4 force field[52] was used as the potential energy function for the protein.

## Target Molecular Dynamics (MD) simulations

We used our cryo-EM closed state structure with ATP and MCC950 bound as a starting point for our MD simulations. The missing loops were built. Then the model was solved with TIP3 water and neutralized with 150 mM KCl using CHARMM-GUI[53], resulting in a box dimension of $130 \times 130 \times 130$ Å$^3$. The default CHARMM-GUI protocol was used to energy minimize the equilibrate the system.

MD simulations in this study were performed using NAMD 2.14[54,55] utilizing CHARMM36m[56] force field parameters. Bonded and short-range nonbonded interactions were calculated every 2 fs, and periodic boundary conditions were employed in all three dimensions. The particle mesh Ewald (PME) method was used to calculate long-range electrostatic interactions every 4 fs with a grid density of 1 Å$^{-3}$. A force-based smoothing function was employed for pairwise nonbonded interactions at a distance of 10 Å with a cutoff of 12 Å. Pairs of atoms whose interactions were evaluated were searched and updated every 20 fs. A cutoff (13.5 Å) slightly longer than the nonbonded cutoff was applied to search for the interacting atom pairs. Constant pressure was maintained at a target of 1 atm using the Nosé–Hoover Langevin piston method. Langevin dynamics maintained a constant temperature of 310 K with a damping coefficient, γ, of 0.5 ps$^{-1}$ applied to all atoms.

The target MD (TMD) simulations were performed by forcing the closed state towards our open state structure using root-mean-square deviation as a collective variable within 50 ns. Given the high force applied, to prevent any unnecessary perturbation harmonic restrains were applied in the LRR region. This region does not undergo any conformational changes relative to the NACHT domain.

The solvent accessible area was calculated using Areaimol within CCP4, based on individual simulation ensembles[49,57].

## ASC Speck assay in HEK293T cells

HEK293T cells (ATCC, Catalog number CRL-3216) were transduced with human ASC-mCherry lentivirus with puromycin selection. Positive cells were selected with 1 µg/ml of puromycin. On the day of the experiment, 10 K cells were plated per well in 96 well plate (Greiner 655090) in the morning. Cells were allowed to be attached for 6 h, cells were transfected with different human NLRP3 constructs using lipofectamine 2000. Next day, around 23 h post transfection, nigericin (500 nM) was added for 2 h. ASC specks (Red counts) were counted using Incucyte S3 (2022B Rev2 software) with basic analyzer using surface-fit analysis.

## Data availability

The coordinates and EM maps generated in this study have been deposited in the Protein Data Bank and the Electron Microscopy Data Bank under accession codes: PDB 8SWK, EMD-40820 (NLRP3$_{\Delta PYD/+ATP/+MCC950}$). PDB 8SXN, EMD-40855 (NEK7/NLRP3$_{\Delta PYD/+ATP/+MCC950}$). PDB 8SWF, EMD-40811 (NLRP3$_{\Delta PYD/+ATP/-MCC950}$). Source data are provided with this paper.

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

## Acknowledgements

We thank Josh Karchin, Adam Huff, Patricia Goode-Mason, Jason Cargill, and Accelagen for reagent support, David Duda, Sandeep Somani, Robert Kirkpatrick, and Ruth Steele for critical data review and discussion of the manuscript. Structural predictions were performed using an internal deployment of AlphaFold2, which was setup and deployed by the In Silico Discovery & External Innovation (ISDEI) team of Johnson & Johnson Innovation Medicine. The cryo-EM data were collected at NanoImaging Services (San Diego, CA).

## Author contributions

X.Y., R.Ma., N.V.O., and S.S. designed the experiments. X.Y., R.Ma., R.Mi., D.C., B.V.S., K.G., J.S., B.P., N.H., Y.Y., G.J.T., L.P., E.L., A.B., D.O., and N.V.O. performed the experiments. X.Y. and R.Ma. analyzed the data, wrote the first draft of the manuscript, and prepared data visualization. X.Y., R.Ma., N.V.O., and S.S. wrote the manuscript. All the authors edited the manuscript.

## Competing interests

The authors declare the following competing interests: X.Y., R.Ma., R.Mi., D.C., B.V.S., K.G., J.S., B.P., Y.Y., G.J.T., L.P., A.B., D.O., N.V.O., and S.S. are employees of Johnson & Johnson Innovative Medicine. The remaining authors declare no competing interests.
