## [Peer Review File · Nature Communications]

Structural basis for the oligomerization-facilitated NLRP3 activationREVIEWER COMMENTS

Reviewer #1 (Remarks to the Author):

In this study, the authors report the cryo-EM structure of PYD-truncated human NLRP3 in the form of an open octamer, which is different from the previously reported cage-shaped inactive form or disk-shaped active form. Considering the structural re-organization of NOD module in the open octamer, the authors propose this is the intermediate state during NLRP3 inflammasome activation and assembly. A 293T cell-based NLRP3 inflammasome activation assay is employed to show that mutations on the open octamer's interaction interface attenuate the IL-1 β secretion.

This is a very interesting body of work, however, there are several points that I would like the authors to consider.

Specific points:

1. Since the determined structure is from the PYD domain truncated NLRP3 protein, whether the full-length NLRP3 could also form the open octamer structure? How to prove that by additional structural or biochemical studies?
2. The authors mention the "self-activation of NLRP3" in the title and the manuscript, however, I can not understand clearly what does that mean. Do the authors mean the NLRP3 can be activated in the absence of any stimuli, or once activated, the NLRP3 could be activated in a self-propagation manner similar to NLRC4? Whatever, I can not find any evidence to support either conclusion in the current manuscript.
3. In the NLRP3 inflammasome activation assay performed in the 293T cell, the authors tried three different doses of transfected NLRP3 plasmid, and measured the IL-1 β secretion without any NLRP3 activation stimuli. Firstly, the ASC speck formation and caspase-1 activation need to be determined among different NLRP3 mutants; Secondly, what is the physiology concentration of NLRP3 in vivo, such as the BMDM cells, and a similar expression level of NLRP3 is expected to perform assay in 293T cells; Last but not least, the key NLRP3 mutations need to be verified in immune cells, such as BMDMs or THP1, and then performed the NLRP3 activation assay in a more physiological condition (LPS+ATP or LPS+Nigericin).
4. What is the relationship between the NEK7 binding and this open octamer formation? Does the NLRP3 expose more NEK7 binding surface in the open octamer compared with the inhibitory cage-shaped structure?
5. What is driving force for NLRP3 to form this open octamer structure from the cage-shaped structure? The hydrolysis of ATP? How the structural re-organization of NOD module leads to the change of oligomeric form of NLRP3?
6. The figure quality need to be improved. For example, what do two different colors mean in Fig. 1B? The whole picture of NLRP3 octamer shown in Fig. 1B should contain multiple side views to make it easier to understand. The label of "one unit" is not clear in Fig. 2A and it is a little hard for readers to understand.

Reviewer #2 (Remarks to the Author):

In the current study, Yu et al. determine the cryo-EM structures of closed and open human NLRP3. Site-directed mutagenesis, computer simulations and cellular assays were used to inform the mechanism of NLRP3 inflammasome self-activation with significant conformational changes in the NACHT domain. Several interfaces (back-back, head-face, tail-tail, and face-face) were identified in the NLRP3 oligomeric state, and key interface sites were mutated sequentially in full-length NLRP3 to understand their role in NLRP3 activation and inflammasome assembly, suggesting their physiological relevance. While the authors used cutting-edge techniques and substantial data analysis, some of the modeling methods applied in the study are not optimal, and additional analyses should be performed.

Major comments

1. The authors should provide a rationale for using a Δ PYD construct rather than full length NLRP3 in their structural analysis, as PYD is required for optimal NLRP3 activity (PMID: 34861190).
2. The authors used an elastic network model based on Brownian dynamics to infer the conformational transitions between open and closed states of NLRP3 by providing the determined structures as inputs. This model is not optimal, as the applied approach does not account for the coordinates of ligands (het atoms of ATP and MCC950) but instead uses Ca atoms of the protein backbone to compute structural transitions and movements. Additionally, the supplementary video demonstrates that the ligands did not experience any fluctuations, but their binding site residues are the first to undergo significant conformational changes. Because MCC950 locks NLRP3 in its closed form and ATP hydrolysis activates NLRP3, these structural transitions and reaction pathways should be studied using an elastic network or coarse-grained models that account for ligand-induced effects. Alternatively, the authors should analyse the dissociation of MCC950/ATP from the closed complex and their subsequent convergence near to the open NLRP3 state using random acceleration-based molecular dynamics (RAMD). This analysis can identify amino acid residues undergoing significant conformational changes that have a high energy barrier.
3. The current study employs mutations in HEK293T cells to understand the mechanisms of NLRP3 activation. However, the authors should validate the mutations they identified in primary or immortalized macrophages using CRISPR and the canonical LPS + ATP trigger. This will significantly strengthen the feasibility of the authors hypotheses.
4. The authors should provide more details about the methods used to generate and apply their models. For instance, the authors should detail the steps involved in NLRP3 octamer modelling using the AlphaFold 2 software. Did they use their own resolved structures as templates to construct the full-length NLRP3 octamer, fill in the unresolved regions (PYD plus linker), and infer the docking site for the α -helix114-128 linker, or was the structure generated from scratch using a standard multimer module? Additionally, the authors should briefly explain how the NLRP3 protomer units were numerically allocated. Furthermore, considering that the solved structure of full-length NLRP3 begins at the 130th amino acid, in Fig. S4B it is unclear how the authors infer that the LRR edge in one of the catalytic pairs (e.g., 4, 4') serves as an additional docking site for the α -helix114-128 linker region. The authors should explain how they predicted the unresolved sections of the receptor NLRP3 protomer (PYD plus linker) occupying the cavities of the catalytic and docking sites, and they should provide a description for how they resolved the α -helix114-128 (Page 5, Line 120), which is unresolved in the reported 3D structure of NLRP3, and discuss its proposed role in the NACHT conformational changes.
5. Currently, F813 was the only amino acid residue representing the back-back interface chosen for mutation and use in functional assays (Fig. 4). It would be worthwhile to mutate the salt bridge residues (D789-R816) to observe their functional effects. It is likely that the salt bridge may have broken in the active form and could have a critical role in addition to F813 in inflammasome formation and/or activation.

Minor Comments

1. The R292 residue should be depicted in Fig. 2C, as R292 is mutated to investigate its role in physiological conditions in the follow up analyses.
2. Similar to the electrostatic complementarity shown in the head-face interface interaction between NBD and adjacent LRR domains (Fig S4B), the authors should create electrostatic surfaces for face-face interactions between LRR domains in high-order forms.
3. For structural comparisons, the authors should reference the previously published PBD structures in the text and figure legends. For instance, these should be included for the NLRC4 structure (Page 6,

Line 147) and the NAIP structure (Fig. 4C caption, Page 16, Line 304).

4. In Fig. 4C, based on structural overlap, the authors referred to the residue (D363) in NLRP3 as the analogous residue to (R288) in NLRC4. However, as seen in the top panel, these residues are four residues apart and do not align well in 3D. Furthermore, they are not grouped together in the same column of the multiple sequence alignment of NLRP3 with human NLRC4. Therefore, these residue pairings should be termed as "close" rather than "corresponding".

5. The authors should include the activated NLRP3 in the structural overlay of open and closed NLRP3 forms in Fig. S7A.

6. The article would benefit from additional proofreading to correct small typos and inconsistencies (ex: "NLRP3 Promoters" should be changed to "NLRP3 protomers" on page 4, line 93; page 5, lines 109, 111, 116; page 14, line 192; in Fig. 5A-B, the components of the NLRP3 complex should be written in subscripts).

Reviewer #3 (Remarks to the Author):

The NLRP3 inflammasome detects signals of cellular damage and initiates inflammatory responses and pyroptotic cell death. This pathway has attracted significant attention in research, and recently several crucial structures are determined, such as the inactive NLRP3/NEK7 complex, the inactive NLRP3 cage, and the active structure of the NLRP3 inflammasome bound to ASC. Despite these advances, the fundamental question of which signal directly activates NLRP3 and the mechanism of NLRP3 activation remain unanswered.

This manuscript by Yu et. al reported a novel open octamer conformation of NLRP3, provides another piece of the picture, and could greatly advance our understanding of the activation mechanisms of NLRP3.

My major question is about the nucleotide in the NLRP3 Δ PYD/+ATP/-MCC950 sample – Did the authors observe any densities corresponding to ATP or ADP in the density map? Alternatively, could they employ HPLC to determine whether ATP underwent hydrolysis?

A related point – It will be helpful if the authors discuss how the preparation of their NLRP3 Δ PYD/+ATP/-MCC950 sample differs from the previously solved NLRP3 cage, which may produce the novel open octamer structure?

And do they see any 2D/3D classes that may correspond to the closed NLRP3 cage in the NLRP3 Δ PYD/+ATP/-MCC950 dataset?

Minor issues:

1. When discussing the residues involved in salt bridge pairs (page 4), a figure of side chain densities of corresponding amino acids should be included as a supplementary figure to validate the accuracy of the model-building process.
2. Line 2 on page 4- It is less clear what the authors mean by "Superpositions of pairs of equivalent subdomains within NACHT further improve the structural similarities and highlight the plasticity within NLRP3."

REVIEWER COMMENTS

Reviewer #1 (Remarks to the Author):

In this study, the authors report the cryo-EM structure of PYD-truncated human NLRP3 in the form of an open octamer, which is different from the previously reported cage-shaped inactive form or disk-shaped active form. Considering the structural re-organization of NOD module in the open octamer, the authors propose this is the intermediate state during NLRP3 inflammasome activation and assembly. A 293T cell-based NLRP3 inflammasome activation assay is employed to show that mutations on the open octamer's interaction interface attenuate the IL-1 β secretion.

This is a very interesting body of work, however, there are several points that I would like the authors to consider.

We greatly thank and appreciate the reviewer's recognition of our work, and for comments to further improve the quality and significance of our study.

Specific points:

1. Since the determined structure is from the PYD domain truncated NLRP3 protein, whether the full-length NLRP3 could also form the open octamer structure? How to prove that by additional structural or biochemical studies?

We purified full-length (FL) NLRP3 without MCC950 using the same method as NLRP3 Δ PYD. The elution profile for full-length NLRP3-MCC950 was similar to NLRP3 Δ PYD/-MCC950, albeit with a slightly broader peak indicating that the FL NLRP3 may exhibit greater heterogeneity compared to the NLRP3 Δ PYD. Cryo-EM studies were attempted on FL NLRP3-MCC950 with and without NEK7. Unfortunately, non-productive aggregations were observed in the images and 2D classification results (attached here), and we did not observe closed cages (as reported previously using FL NLRP3 with MCC950, PDB ID: 7LFH, and 7VTP), the activated NLRP3 inflammasome disc (as reported previously using co-expression of FL NLRP3, NEK7, and ASC, PDB ID:8EJ4), or the open cage formed by NLRP3 Δ PYD/-MCC950.

We also conducted native mass spectrum studies on the purified NLRP3 Δ PYD/+ATP/-MCC950 without any clear results. From the initial 2D classification on the NLRP3 Δ PYD/+ATP/-MCC950 dataset (attached here), we observed significant particle heterogeneity, including small and large complexes. NLRP3 proves challenging for obtaining a homogenous sample for structural studies. Removing the PYD domain or using conformational restraint inhibitors like MCC950 is essential to improve homogeneity and help to capture snapshots of NLRP3 activation steps.

We conducted molecular dynamic simulations on the open FL NLRP3 model (the Alphafold2 PYD domain model was introduced to the NLRP3 Δ PYD/+ATP/-MCC950 open cage), confirming its stability within 100 ns (Movie 3). The significance of the interfaces observed in the open cage structures was subsequently validated in our cell model. The authors plan to perform cryo-tomography on NLRP3-primed cell lamellae to investigate the presence of closed, open cages, or the activated NLRP3 inflammasome in the future.

D Initial 2D classification results for the NLRP3 Δ PYD/+ATP/-MCC950 dataset

2. The authors mention the “self-activation of NLRP3” in the title and the manuscript, however, I can not understand clearly what does that mean. Do the authors mean the NLRP3 can be activated in the absence of any stimuli, or once activated, the NLRP3 could be activated in a self-propagation manner similar to NLRC4? Whatever, I can not find any evidence to support either conclusion in the current manuscript.

Thank you for the suggestion.

Title change “Structural basis for the oligomerization-facilitated NLRP3 activation”

3. In the NLRP3 inflammasome activation assay performed in the 293T cell, the authors tried three different doses of transfected NLRP3 plasmid, and measured the IL-1 β secretion without any NLRP3 activation stimuli. Firstly, the ASC speck formation and caspase-1 activation need to be determined among different NLRP3 mutants; Secondly, what is the physiology concentration of NLRP3 in vivo, such as the BMDM cells, and a similar expression level of NLRP3 is expected to perform assay in 293T cells; Last but not least, the key NLRP3 mutations need to be verified in immune cells, such as BMDMs or THP1, and then performed the NLRP3 activation assay in a more physiological condition (LPS+ATP or LPS+Nigericin).

Thank you for the suggestion and the authors agree that using immune cells such as BMDMs or THP1 would be more relevant.

The 293T cell assay, as established and utilized by others (Shi et al. 2015, Bio Protoc; Shi et al. 2016, Nat Immunol; Vande Walle et al. 2019, PNAS), offers distinct advantages. It provides a relatively clean and robust environment for assessing the impact of oligomerization on NLRP3 activation. In contrast, BMDMs or THP1 systems are more complex, involving both canonical and non-canonical NLRP3 activation pathways. Furthermore, LPS activation has been shown to induce canonical, non-canonical, and alternative inflammasome activation (Yang Yang et al. 2019 Cell Death & Disease), making it challenging to decipher the specific effects of oligomerization on NLRP3 activation. Our cellular assay further demonstrated that Nigericin more effectively induced NLRP3 activation and subsequent IL-1 β signaling (Fig. S8C), implying that Nigericin may promote NLRP3 activation independently of oligomeric states.

Indeed, we acknowledge the limitations of the 293T cell assay, such as potential differences in NLRP3 expression compared to BMDM/THP1 cells after priming. To address this concern, we conducted experiments using various doses of transfected NLRP3 plasmid. Mutations on the open octamer interfaces do impact IL-1 β signaling, underscoring the biological relevance of the open octamer in the NLRP3 activation pathway. Our team has been actively working on establishing an internal human immune cell-based system for these mutants. However, we have not yet achieved a successful platform. Once the platform is established, we plan to evaluate the mutation data, explore trigger activation, investigate the crosstalk between NLRP3 and NLRC4 activation, conduct cryo-ET studies on ASC species in the cell, and more. We are continuing this work, but it may take a considerable amount of time to achieve success, which could result in a delay in releasing our structural analysis findings to the field. Given the time sensitivity and the broader scientific community, we would like to share our current results with the field, as they can inspire other researchers to design cellular assays that can effectively validate our proposed models.

4. What is the relationship between the NEK7 binding and this open octamer formation? Does the NLRP3 expose more NEK7 binding surface in the open octamer compared with the inhibitory cage-shaped structure?

The Face-Face interaction within the LRRs is of potential interest to readers. We have included a discussion on this topic in the main manuscript, along with two supplementary figures and one table.

In the main text:

“The calculated NEK7 binding interface with the closed NLRP3 $_{\Delta\text{PYD}/+\text{ATP}/+\text{MCC950}}$ measures approximately 2249.2 Å², involving around 74 contact residues (Fig. S6BC, Table S2). In the closed NLRP3 $_{\Delta\text{PYD}/+\text{ATP}/+\text{MCC950}}$ hexamer, the NEK7 binding site at LRR is open accessible. A structural overlay using the protomers from the closed NLRP3 $_{\Delta\text{PYD}/+\text{ATP}/+\text{MCC950}}$ hexamer and NEK7/NLRP3 $_{\Delta\text{PYD}/+\text{ATP}/+\text{MCC950}}$ dimer reveals that the N-lobe of NEK7 clashes with the adjacent NLRP3 protomer within the closed NLRP3 $_{\Delta\text{PYD}/+\text{ATP}/+\text{MCC950}}$ hexamer, leading to the closed hexamer dissociation. In the closed FL NLRP3 cages, the NEK7 binding interface is shielded by the Face-Face interaction (Fig. S6BCD). However, both the calculated Face-Face interface and the contact residues are smaller than those of the NEK7/NLRP3 binding interface (Table S2). In the NLRP3 $_{\Delta\text{PYD}/+\text{ATP}/+\text{MCC950}}$ open octamer structure, the NEK7 binding surface can be occupied by either the neighboring protomer's LRR resulting in a Face-Face interaction or the neighboring protomer's NACHT forming a Head-Face interaction (Fig. 2CE, 3AD). Both the calculated area and contact residues at the Face-Face and Head-Face interfaces are approximately half the values of the NEK7/NLRP3 binding interface (Fig. S6BC, Table S2). Docking NEK7 at the LRR binding sites in NLRP3 closed cages or open octamer results in clashes with the neighboring protomer (Fig. 5D, S6D). These findings suggest that the presence of NEK7 leads to the dissociation of large NLRP3 oligomers, both closed and open, into NEK7/NLRP3 dimers, while the Back-Back interaction is maintained.”

In the discussion:

“NEK7 binds to the concave side of LRR, which largely overlaps with the (Head)Face-Face interactions in the open octamer or Face-Face interactions in the closed cages (Fig. 5D, S6D). Based on interface area and residue count (Table S2), NEK7 binding to the LRR site can probably dissociate both closed cages and open octamers. Hence, the presence of NEK7 should be tightly regulated to interact with the open octamer, to avoid dissociation of the closed cage, resulting in non-productive closed NEK7/NLRP3 (Fig. 6, S13B).”

Fig. S6. Structural analysis of the LRR concave side in closed and open NLRP3 oligomers. **A.** Structural overlays of LRRs from FL mouse (green, PDB ID: 7LFH) and human (cyan, PDB ID: 7PZC) closed cages (top) and human NLRP3 Δ PYD/+ATP/-MCC950 open octamer (yellow, this study, bottom). Angular deviations between human and mouse closed cages and between human closed and open cages were illustrated with curved arrows. **B.** Surface charge distribution patterns of LRR concave side surfaces from the mouse, human FL NLRP3 closed cages, open NLRP3 Δ PYD/+ATP/-MCC950 octamer, and NEK7/NLRP3 Δ PYD/+ATP/-MCC950 dimer structures. **C.** Top to bottom surface representations of NLRP3 in the NEK7 binding, mouse and human closed cages, and open octamer states with concave interaction residues highlighted in orange, red, magenta, green, and cyan. NACHT and LRR were in slate and gray, respectively. **D.** From top to bottom, docking NEK7 onto the LRR concave side in mouse, human closed cages, and open octamer states.

Table. S2. The interface area and contact residues on the concave side of the LRR for the NEK7/NLRP3 dimer (this study), mouse closed cage (PDB ID: 7LFH), human closed cage (PDB ID 7PZC), and open octamer (this study) were calculated using PDBePISA server.

Interface	NLRP3-NEK7 (This study)	Mus Closed cage (PDB ID: 7LFH)	Human Closed cage (PDB ID: 7PZC)		Human Open cage (This study)	
Contact	NEK7-LRR	Face-Face	Face-Face	Acidic loop-Face	Total	Face-Face Head-Face
Interface area (Å ²)	~2249.2	~1049.2	~724.0	~1129.6	~1853.6	~1207.1 ~757.2
Contact Residues	74	35	22	42	64	36 26
Overlap of Contact Residues with NEK7 Binding	74	19	9	24	33	15 23

5. What is driving force for NLRP3 to form this open octamer structure from the cage-shaped structure? The hydrolysis of ATP? How the structural re-organization of NOD module leads to the change of oligomeric form of NLRP3?

In the discussion, we've outlined various hypotheses regarding NLRP3 oligomeric reorganization and activation. Please refer to the attached main text for a detailed summary.

1. Comparing human/mouse closed cages with the open octamer reveals Face-Face interface plasticity and relatively stable Back-Back interactions (Fig. 3B, S6AB).

2. ATP hydrolysis potentially induces NACHT perturbations enabling conformational changes.

3. The open octamer provides an LRR scaffold with a large negatively charged surface, attracting the positively charged NACHT-NBD interface, facilitating NACHT conformational changes (Fig. 3D, S5).

4. The NACHT opening and cage rearrangement from closed to open may occur simultaneously or mutually support one another.

5. Altering properties or position of the acidic-loop₆₈₉₋₆₉₈ might aid NACHT opening.

In the main text, discussion:

“Based on available cryo-EM structures, we propose an activation mechanism for NLRP3 (Fig. 6). This involves a two-step process starting with priming, which triggers the NLRP3 transcriptional upregulation and oligomerization forming the closed cages. The formation of the closed NLRP3 cages was driven by LRR-mediated repeated Face-Face and Back-Back interactions, while NACHTs do not directly interact (Fig. 3C) 5,6. Structural comparison between human and mouse closed cages shows Face-Face interface plasticity and relatively stable Back-Back interactions (Fig. 3AB, S6AB). The closed NLRP3 cages can be activated by a second signal, such as ATP (Fig. S1C). From the closed NLRP3 structures, ATP forms interactions with NBD, HD1, and WHD (Fig. 5C). ATP hydrolysis potentially induces NACHT perturbations enabling conformational changes. Additionally, the open octamer provides an LRR scaffold with a large negatively charged surface, attracting the positively charged NACHT-NBD interface, facilitating NACHT conformational changes (Fig. 3D, S5). The NACHT opening and cage rearrangement from closed to open may occur simultaneously or mutually support one another (Movie 2, Fig. 3D). Note that the PYD is not involved in this proposed NLRP3 activation mechanism. In the closed FL NLRP3 cages, PYDs were either found at the center of cages or exhibited dynamic movement around the cages. In the open octamer, MD simulation results indicate that all PYDs surround the octamer, following the N-terminal trend of NBD (Fig. S7A, Movie 3). This suggests that PYD does not impact octamer formation; instead, it may be located at a binding site on NBD, which is only present in the open and activated forms (Fig. S7, Movie 3).”

“The acidic-loop₆₈₉₋₆₉₈ was found to shield the LRR-catalytic surface in the closed human cage (PDB ID: 7PZC) and was not resolved in the open octamer (Fig. S13A). The acidic loop can hinder receptor-NBD binding to catalytic-LRR concave surface (Fig. S6C). It can also impede NEK7 binding to the LRR. Indeed, deletion of the acidic loop showed a marked loss of autoinhibition, facilitating NLRP3 activation. However, we cannot eliminate the possibility that altering properties or position of the acidic loop might aid NACHT opening. Two GOF mutations in this acidic-loop region were linked to NLRP3 constitutive activation (Fig. S9). NEK7 engagement or Face-Face, and Head-Face interactions that were observed in the open octamer may help reposition the acidic loop to aid NACHT opening (Fig. S13A). Indeed, the binding of NEK7 altering the acidic loop slightly opened the NACHT domain even in the presence of MCC950 (Fig. S11, S13A).”

6. The figure quality need to be improved. For example, what do two different colors mean in Fig. 1B? The whole picture of NLRP3 octamer shown in Fig. 1B should contain multiple side views to make it easier to understand. The label of “one unit” is not clear in Fig. 2A and it is a little hard for readers to understand.

We agree with the reviewer’s suggestion to include varied views of the cryo-EM map of NLRP3_{ΔPYD/+ATP/-MCC950} were provided in Fig. 1B. The NACHT and LRR domains were in slate and gray, respectively, with the key interfaces between protomers highlighted.

“We’ve removed the label ‘one unit’ from Fig. 2A to avoid reader confusion. Instead, we believe it’s more suitable to introduce the concept of ‘one unit’ in Fig. 3B. In the main text, the following sentence is introduced on Page 5.” The closed FL NLRP3 cages formed through repetitive Back-Back and Face-Face interactions among the LRRs. For simplicity in the following discussion, we’ll refer to the NLRP3 dimers with

Back-Back interactions as one unit, and four consecutive units with Face-Face interactions were used to elucidate the activation mechanism (Fig. 3A)."

Fig. 1. Cryo-EM structure of open NLRP3_{ΔPYD/+ATP/-MCC950} octamer

A. Domain organization of human NLRP3 with PYD domain truncated. **B.** Different views of cryo-EM map of NLRP3_{ΔPYD/+ATP/-MCC950}. The NACHT and LRR domains were in slate and gray, respectively, with the key interfaces between protomers highlighted. One NLRP_{ΔPYD} protomer was highlighted. 2D classes from cryo-EM images of the sample were shown at the bottom. **C.** Structural overlay of open NLRP3_{ΔPYD/+ATP/-MCC950} to the closed NLRP3 (Left) or activated NEK7/NLRP3 (Right). The LRR region and NEK7 were colored in gray and orange, respectively. The NACHT domains were in cyan, slate, or green for the closed, open, or activated states, respectively. **D.** Zoom-in views of NACHT conformational changes. NBD and HD1 were in orange and cyan or magenta and slate for the closed or open states, respectively. WHD and HD2 were in light and dark green, respectively. The nucleotide and MCC950 from the closed NLRP3 were shown as sticks. The protein structures were shown as cartoon. Subdomain reorganization upon activation was highlighted by arrows.

respectively. **D.** Zoom-in views of NACHT conformational changes. NBD and HD1 were in orange and cyan or magenta and slate for the closed or open states, respectively. WHD and HD2 were in light and dark green, respectively. The nucleotide and MCC950 from the closed NLRP3 were shown as sticks. The protein structures were shown as cartoon. Subdomain reorganization upon activation was highlighted by arrows.

Fig. 3. Purposed NLRP3 activation mechanism.

A. Schematic representation of the arrangement of the oligomerization transition from the closed to the open cages. NACHT was not shown. The Back-Back, Face-Face, Head-Face, and Tail-Tail interactions were highlighted using magenta, gray, pink, and yellow spheres, respectively. NLRP3 dimers with Back-Back interactions was highlighted as one unit. **B.** Structural overlays of NLRP3 closed cage (PDB ID: 7LFH) and open octamer (this study) at face-face interface using one protomer as the reference. NACHT were colored in slate. LRR were in red or gray for closed cage or open octamer, respectively. **C.** Structural overlays of NLRP3 closed (red, PDB ID: 7LFH) and open (slate and gray, this study) cages. LRRs were highlighted. **D.** Schematic representation of the proposed activation mechanism at the NACHT domain in the octamer. Note, two consecutive units were shown.

(slate and gray, this study) cages. LRRs were highlighted. **D.** Schematic representation of the proposed activation mechanism at the NACHT domain in the octamer. Note, two consecutive units were shown.

Reviewer #2 (Remarks to the Author):

In the current study, Yu et al. determine the cryo-EM structures of closed and open human NLRP3. Site-directed mutagenesis, computer simulations and cellular assays were used to inform the mechanism of NLRP3 inflammasome self-activation with significant conformational changes in the NACHT domain. Several interfaces (back-back, head-face, tail-tail, and face-face) were identified in the NLRP3 oligomeric state, and key interface sites were mutated sequentially in full-length NLRP3 to understand their role in NLRP3 activation and inflammasome assembly, suggesting their physiological relevance. While the authors used cutting-edge techniques and substantial data analysis, some of the modeling methods applied in the study are not optimal, and additional analyses should be performed.

We sincerely appreciate the reviewer's valuable feedback for enhancing the quality and significance of our study. Below, we provide specific responses to the reviewer's comments on our manuscript regarding the novel open octamer NLRP3 structure.

Major comments

1. The authors should provide a rationale for using a Δ PYD construct rather than full length NLRP3 in their structural analysis, as PYD is required for optimal NLRP3 activity (PMID: 34861190).

The authors agree with the reviewer regarding the significance of PYD in NLRP3 activation and inflammasome assembly. PYD's presence contributes to the formation of closed cages (PMID: 34861190). Unlike NLRC4, a disk-shaped NLRP3 structure has only been observed in cases of co-expression of FL NLRP3 with NEK7, followed by ASC PYD incubation (PMID: 36442502). Our studies using NLRP3 Δ PYD/+ATP/-MCC950 demonstrate that conformational changes, a critical step in NLRP3 activation, can occur without the need for PYD or ASC. The open NLRP3 form exposes a significant catalytic surface for inflammasome assembly, but unsupervised mixing of open NEK7/NLRP3 can lead to aggregation. PYD domains and ASC play roles in forming helical conformations, guiding inflammasome assembly.

We attempted multiple experiments using FL NLRP3_{+ATP/-MCC950} with/without NEK7. The size exclusion profile of FL NLRP3_{+ATP/-MCC950} showed a broader peak compared to NLRP3 Δ PYD/+ATP/-MCC950 indicating that the FL NLRP3 may exhibit greater heterogeneity compared to the NLRP3 Δ PYD. EM micrographs showed non-productive aggregation, with no closed cages, open octamer, or disk-shaped NLRP3 feature particles observed in the 2D classification. Obtaining a homogenous sample for structural studies with NLRP3 is challenging. Removing the PYD domain or using conformational restraint inhibitors like MCC950 is essential to improve homogeneity and facilitate the capture of NLRP3 activation snapshots.

2. The authors used an elastic network model based on Brownian dynamics to infer the conformational transitions between open and closed states of NLRP3 by providing the determined structures as inputs. This model is not optimal, as the applied approach does not account for the coordinates of ligands (het atoms of ATP and MCC950) but instead uses C α atoms of the protein backbone to compute structural transitions and movements. Additionally, the supplementary video demonstrates that the ligands did not experience any fluctuations, but their binding site residues are the first to undergo significant conformational changes. Because MCC950 locks NLRP3 in its closed form and ATP hydrolysis activates NLRP3, these structural transitions and reaction pathways should be studied using an elastic network or coarse-grained models that account for ligand-induced effects. Alternatively, the authors should analyse the dissociation of MCC950/ATP from the closed complex and their subsequent convergence near to the open NLRP3 state using random acceleration-based molecular dynamics (RAMD). This analysis can identify amino acid residues undergoing significant conformational changes that have a high energy barrier.

We do agree with the reviewer's comment that the elastic network model based on Brownian dynamics is not an optimal way of investigating protein conformational transitions. By following the suggestion, we have now performed a completely new set of all-atom molecular dynamics (MD) simulations, where we use the closed state model bound with ATP and MCC950 inhibitor as the starting model, solvated with water and neutralized with ions (more details on the method section of the revised manuscript), and drive the conformation towards the open state using targeted molecular dynamics (TMD) simulations. We chose to use this method because of our previous experience. While TMD itself is a non-equilibrium method, meaning the time at which we simulated the process is much faster than the real biological time scale, we indeed observed that the inhibitor leaves the binding site when it reaches the open state, as expected. To get a better picture of the dynamic process, we will need much more exhaustive simulations such as enhanced sampling or Markov state modeling. However, this is beyond the scope of this study and will be reported in a follow-up study. Meanwhile, we have updated Fig. S12 (attached here) and provided a movie 4 based on the new results.

3. The current study employs mutations in HEK293T cells to understand the mechanisms of NLRP3 activation. However, the authors should validate the mutations they identified in primary or immortalized macrophages using CRISPR and the canonical LPS + ATP trigger. This will significantly strengthen the feasibility of the authors hypotheses.

Thank you for the suggestions, especially regarding immortalized macrophages using CRISPR. While we actively strive to establish an internal human immune cell-based system for these mutants, we have not yet achieved success. Our future plans involve assessing mutation data, investigating trigger activation, exploring the interaction between NLRP3 and NLRC4 activation, conducting cryo-ET studies on ASC species within cells, and more. Although we are making progress, the timeline for success remains uncertain, potentially causing a delay in releasing our structural analysis findings to the scientific community. Acknowledging the time-sensitive nature and broader scientific interest, we aim to share our current results to inspire other researchers in designing effective cellular assays for validating our proposed models.

The 293T cell assay, as established and utilized by others (Shi et al. 2015, *Bio Protoc*; Shi et al. 2016, *Nat Immunol*; Vande Walle et al. 2019, *PNAS*), offers distinct advantages. It provides a relatively clean and robust environment for assessing the impact of oligomerization on NLRP3 activation. In contrast, BMDMs or THP1 systems are more complex, involving both canonical and non-canonical NLRP3 activation pathways. Furthermore, LPS activation has been shown to induce canonical, non-canonical, and alternative inflammasome activation (Yang Yang et al. 2019 *Cell Death & Disease*), making it challenging to decipher the specific effects of oligomerization on NLRP3 activation. Our cellular assay further demonstrated that Nigericin more effectively induced NLRP3 activation and subsequent IL-1 β signaling (Fig. S8C), implying that Nigericin may promote NLRP3 activation independently of oligomeric states.

4. The authors should provide more details about the methods used to generate and apply their models. For instance, the authors should detail the steps involved in NLRP3 octamer modelling using the AlphaFold 2 software. Did they use their own resolved structures as templates to construct the full-length NLRP3 octamer, fill in the unresolved regions (PYD plus linker), and infer the docking site for the α -helix114-128 linker, or was the structure generated from scratch using a standard multimer module? Additionally, the authors should briefly explain how the NLRP3 protomer units were numerically allocated. Furthermore, considering that the solved structure of full-length NLRP3 begins at the 130th amino acid, in Fig. S4B it is unclear how the authors infer that the LRR edge in one of the catalytic pairs (e.g., 4, 4') serves as an additional docking site for the α -helix114-128 linker region. The authors should explain how they predicted the unresolved sections of the receptor NLRP3 protomer (PYD plus linker) occupying the cavities of the catalytic and docking sites, and they should provide a description for how they resolved the α -helix114-128 (Page 5, Line 120), which is unresolved in the reported 3D structure of NLRP3, and discuss its proposed role in the NACHT conformational changes.

In the method part:

Model building and refinement

“The EM density map of the open octamer reveals distinct features corresponding to the LRR, while the closed form NLRP3 structure only allows for LRR docking, with the NACHT domain extending beyond the map. Furthermore, during the structural determination and modeling of the NLRP3 open octamer, no activated form structure was available as an initial model. An Alphafold2 (AF2) FL NLRP3 model (AF-Q96P20-F1) was used as an initial model. Individual subdomain structures (NBD, HD1, WHD, HD2, and LRR) were extracted from the AF2 model and then docked onto one protomer of the open octamer EM density map using Chimera 7. These docked subdomains were merged into a single protomer, removing the disordering loops without the EM density support, and followed by manual adjustment using COOT 8. The complete NLRP3 octamer structure was created from a single protomer and then expanded into a symmetric structure (D2) using Chimera software.”

Molecular Dynamics (MD) simulation

“To assess the impact of PYD domains on the open NLRP3 octamer, we conducted molecular dynamic (MD) simulations using a chimera model derived from the open NLRP3_{ΔPYD} structure and the PYD subdomain model from the AF2 FL NLRP3 model (AF-Q96P20-F1). The PYD₁₋₁₄₇ model was isolated from the AF2 model and fused to each individual protomer within the open octamer using the α -helix₁₃₆₋₁₄₇ as the reference. The resulting chimera FL NLRP3 open octamer was subjected to the default relaxation protocol in Maestro and followed by a 100 ns MD simulation. The simulation was run at 300 K and 1 bar for 100 ns using the Desmond simulation package in Schrodinger 2021-1 (www.schrodinger.com). The systems were protonated at neutral pH and centered in a cubic box such that the minimum distance from any protein atom to the box wall was 10 Å. The box was solvated using SPC 18 water molecules and counter ions were added to neutralize the system. OPLS4 force field was used as the potential energy function for the protein.”

In the main text:

“To assess the impact of PYD domains on the open NLRP3 octamer, we conducted molecular dynamic (MD) simulations using a chimera model derived from the open NLRP3_{ΔPYD} structure and the PYD subdomain model from the AlphaFold2 (AF2) FL NLRP3 model (AF-Q96P20-F1). The PYD₁₋₁₄₇ (amino acid (a.a.), 1-147) model was isolated from the human FL NLRP3 AF2 model and fused to each individual protomer within the open octamer using the α -helix₁₃₆₋₁₄₇ as the reference. The resulting chimera FL NLRP3 open octamer was subjected to the default relaxation protocol in Maestro and followed by a 100 ns MD simulation (Movie 3). The FL NLRP3 open octamer model maintained stability throughout the MD simulation, with consistent Back-Back, Head-Face, Face-Face, and Tail-Tail interactions. Notably, the modeled PYDs exhibited relatively stable interactions at the NBD's rear, forming Head-Face interactions (Fig. S7A). This putative PYD docking site was exclusively present in the open and activated NLRP3 form, partially obscured by the WHD domain in the closed conformation (Fig. S7B). At the edge of catalytic LRR, one positively charged patch could serve as an extra docking site for the predominantly negatively charged α -helix₁₁₄₋₁₂₈ from the receptor NLRP3 protomer (Fig. S7C), which can assist the conformational changes on the NACHT, meanwhile positioning the PYD domain.”

Fig. S7. Proposed PYD position in NLRP3_{ΔPYD}/+ATP/-MCC950 octamer. A. NLRP3_{ΔPYD}/+ATP/-MCC950 structure and FL AF2 NLRP3 chimera model in the octamer states were shown as cartoon with the PYD, NACHT, and LRR in red, slate, and gray, respectively. B. Putative PYD binding site on the open/activated NLRP3. Left, structural overlay of closed NLRP3_{ΔPYD}/+ATP/-MCC950 and open PYD-NLRP3_{ΔPYD}/+ATP/-MCC950 chimera model after MD simulation, using NBD domain as a reference. The dashed circle indicated the clashes between the PYD model and closed NACHT (in cyan). Right, structural overlay of activated FL NLRP3 (PDB ID: 8EJ4) and open PYD-NLRP3_{ΔPYD}/+ATP/-MCC950 chimera model after MD simulation, using NBD domain as a reference. The NACHT in closed, open, and activated NLRP3 is colored cyan, slate, and green, respectively, while the PYD and LRR are red and gray, respectively. C. Proposed docking site for the α -helix₁₁₄₋₁₂₈ on the catalytic LRR in the open octamer state. Surface charge distribution

pattern of the catalytic protomer (top panel), and α -helix₁₁₄₋₁₂₈ (lower panel) were shown with the counterparts shown in cartoon. The sequence of α -helix₁₁₄₋₁₂₈ was shown above.

Regarding how the NLRP3 protomer units were numerically allocated

In the main text, the following sentence is introduced on Page 5. "The closed FL NLRP3 cages formed through repetitive Back-Back and Face-Face interactions among the LRRs. For simplicity in the following discussion, we'll refer to the NLRP3 dimers with Back-Back interactions as one unit, and four consecutive units with Face-Face interactions were used to elucidate the activation mechanism (Fig. 3A)."

Fig. 3. Purposed NLRP3 activation mechanism.

A. Schematic representation of the arrangement of the oligomerization transition from the closed to the open cages. NACTH was not shown. The Back-Back, Face-Face, Head-Face, and Tail-Tail interactions were highlighted using magenta, gray, pink, and yellow spheres, respectively. NLRP3 dimers with Back-Back interactions was highlighted as one unit. **B.** Structural overlays of NLRP3 closed cage (PDB ID: 7LFH) and open octamer (this study) at face-face interface using one protomer as the reference. NACTH were colored in slate. LRR were in red or gray for closed cage or open octamer, respectively. **C.** Structural overlays of NLRP3 closed (red, PDB ID: 7LFH) and open (slate and gray, this study) cages. LRRs were highlighted. **D.** Schematic representation of the proposed activation mechanism at the NACTH domain in the octamer. Note, two consecutive units were shown.

at the NACTH domain in the octamer. Note, two consecutive units were shown.

5. Currently, F813 was the only amino acid residue representing the back-back interface chosen for mutation and use in functional assays (Fig. 4). It would be worthwhile to mutate the salt bridge residues (D789-R816) to observe their functional effects. It is likely that the salt bridge may have broken in the active form and could have a critical role in addition to F813 in inflammasome formation and/or activation.

Thank you for the suggestion. We introduced additional substitutions at the Back-Back interface. Surprisingly, these multiple substitutions did not enhance IL-1 β signaling; instead, they resulted in a progressive decrease in signaling (Fig. S7C, attached here), supporting the reviewer's hypothesis.

In the main text discussion part:

"The Back-Back interaction has a dual effect on the NLRP3 activation and inflammasome assembly. It is observed in the closed and open NLRP3 oligomers but absent in the activated NLRP3 inflammasome (Fig. 2AB, 3). They are necessary for NLRP3 oligomer formation but the observed NEK7/NLRP3 dimer, facilitated by Back-Back interaction, can also impede NLRP3 inflammasome assembly by covering the receptor surface for subsequent NEK7/NLRP3 recruitment (Fig. 5B, S13B). The NEK7/NLRP3 most likely exists in a dynamic equilibrium between monomers 3 (PDB ID: 6NPY) and dimers (this study) (Fig. 6, S13B), which allows the closed cages with even-numbered protomers (6-, 8-, 10-, 12-, 14-, or 16- mers) 4-6 to eventually form the

disk-shaped inflammasomes with even or odd- numbered protomers (10 or 11- mers NLRP3 inflammasome complex) (Fig. 6, Fig. S13B). A point mutation (Phe813Asp) disrupting the Back-Back interaction shifts the balance towards the open NEK7/NLRP3 monomers and enhances IL-1 β signaling in a cell model (Fig. 4B). Additional substitutions at the Back-Back interface (Fig. S8C) progressively decreased IL-1 β levels, suggesting they impact both closed and open cage formation and oligomerization-facilitated NLRP3 activation.”

Minor Comments

1. The R292 residue should be depicted in Fig. 2C, as R292 is mutated to investigate its role in physiological conditions in the follow up analyses.

Thank you for the suggestion. Fig. 2C has been updated.

2. Similar to the electrostatic complementarity shown in the head-face interface interaction between NBD and adjacent LRR domains (Fig S4B), the authors should create electrostatic surfaces for face-face interactions between LRR domains in high-order forms.

Thanks for the valuable suggestion. The authors include an extra supplementary figure to clarify the structural information for the audience.

Fig. S6. Structural analysis of the LRR concave side in closed and open NLRP3 oligomers. A. Structural overlays of LRRs from FL mouse (green, PDB ID: 7LFH) and human (cyan, PDB ID: 7PZC) closed cages (top) and human NLRP3 Δ PYD/+ATP/-MCC950 open octamer (yellow, this study, bottom). Angular deviations between human and mouse closed cages and between human closed and open cages were illustrated with curved arrows. B. Surface charge distribution patterns of LRR concave side surfaces from the mouse, human FL NLRP3 closed cages, open NLRP3 Δ PYD/+ATP/-MCC950 octamer, and NEK7/NLRP3 Δ PYD/+ATP/-MCC950 dimer structures. C. Top to bottom surface representations of NLRP3 in the NEK7 binding, mouse and human closed cages, and open octamer states with concave interaction residues highlighted in orange, red, magenta, green, and cyan. NACTH and LRR were in slate and gray, respectively. D. From top to bottom, docking NEK7 onto the LRR concave side in mouse, human closed cages, and open octamer states.

3. For structural comparisons, the authors should reference the previously published PDB structures in the text and figure legends. For instance, these should be included for the NLRC4 structure (Page 6, Line 147) and the NAIP structure (Fig. 4C caption, Page 16, Line 304).

PDB IDs have been updated in the figure legends.

4. In Fig. 4C, based on structural overlap, the authors referred to the residue (D363) in NLRP3 as the analogous residue to (R288) in NLRC4. However, as seen in the top panel, these residues are four residues apart and do not align well in 3D. Furthermore, they are not grouped together in the same column of the multiple sequence alignment of NLRP3 with human NLRC4. Therefore, these residue pairings should be termed as "close" rather than "corresponding".

Thank you for the suggestion. The word has been changed to "close".

5. The authors should include the activated NLRP3 in the structural overlay of open and closed NLRP3 forms in Fig. S7A.

Thank you for the suggestion. The activated NLRP3 structure has been introduced to both the S10 (old Fig. S7) and S11 figures.

Fig. S10

Fig. S11

6. The article would benefit from additional proofreading to correct small typos and inconsistencies (ex: “NLRP3 Promoters” should be changed to “NLRP3 protomers” on page 4, line 93; page 5, lines 109, 111, 116; page 14, line 192; in Fig. 5A-B, the components of the NLRP3 complex should be written in subscripts).

Thank you, the reviewer, for carefully checking. The typos have been corrected. Figure 5 has been updated and copied here.

Fig. 5. Cryo-EM structures of NLRP3_{ΔPYD/+ATP/+MCC950} in complex with NEK7.

A. B. Cryo-EM maps of closed NLRP3_{ΔPYD/+ATP/+MCC950} hexamer and NEK7/NLRP3_{ΔPYD/+ATP/+MCC950} dimer, respectively. Domain organization of NEK7 was shown. The NACT, LRR, NEK7-N-lobe, and NEK7-C-lobe were colored in slate, gray, red, and orange, respectively. One protomer was highlighted, respectively. **C.** Structure of closed NLRP3_{ΔPYD/+ATP/+MCC950} protomer within the hexamer (Left) and structural overlay of closed NLRP3_{ΔPYD/+ATP/+MCC950} hexamer and NEK7/NLRP3_{ΔPYD/+ATP/+MCC950} dimer at MCC950 binding site (Right). ATP and MCC950 were shown as sticks. NACT subdomains NBD, HD1, WHD, and HD2 were colored in orange, slate, light green, and dark green, respectively. Loop₆₁₇₋₆₂₈ was highlighted. **D.** From top to down, structure of NEK7/NLRP3_{ΔPYD/+ATP/+MCC950} dimer, docking of NEK7 to the concave sides of LRRs in the closed hexamer (this study), FL cage (PDB ID: 7LFH) and open octamer (this study). The Back-Back, Face-Face, Head-Face, and Tail-Tail interactions were highlighted using magenta, gray, pink, and yellow spheres, respectively. The NACT, LRR, and NEK7 were colored in slate, gray, and orange, respectively.

Reviewer #3 (Remarks to the Author):

The NLRP3 inflammasome detects signals of cellular damage and initiates inflammatory responses and pyroptotic cell death. This pathway has attracted significant attention in research, and recently several crucial structures are determined, such as the inactive NLRP3/NEK7 complex, the inactive NLRP3 cage, and the active structure of the NLRP3 inflammasome bound to ASC. Despite these advances, the fundamental question of which signal directly activates NLRP3 and the mechanism of NLRP3 activation remain unanswered.

This manuscript by Yu et. al reported a novel open octamer conformation of NLRP3, provides another piece of the picture, and could greatly advance our understanding of the activation mechanisms of NLRP3.

We appreciate the reviewer's valuable feedback and constructive comments on our manuscript regarding the novel open octamer NLRP3 structure. Below please see our specific answers to these comments:

My major question is about the nucleotide in the NLRP3 Δ PYD/+ATP/-MCC950 sample – Did the authors observe any densities corresponding to ATP or ADP in the density map? Alternatively, could they employ HPLC to determine whether ATP underwent hydrolysis?

We did not use HPLC for ATP hydrolysis determination. Instead, we conducted the NLRP3 ATPase activity and inhibition assay using purified recombinant human NLRP3 Δ PYD, the same construct as in our structural studies. (NLRP3 ATPase activity and inhibition in Method) Results show NLRP3 Δ PYD has ATPase activity, with MCC950 significantly inhibiting it. Cryo-EM grids were prepared within 30 minutes of sample purification.

The ATP hydrolysis profile suggests a mixture of both ATP and ADP at the grid preparation time. In addition, initial 2D classification (attached here) revealed sample heterogeneity with small and large particles. Therefore, HPLC may not yield a definitive result regarding ATP/ADP engagement with the open NLRP3 octamer.

D

Our cryo-EM map of the NLRP3 open octamer did not reveal additional density corresponding to ATP/ADP. The open octamer's nucleotide-binding pocket is partially solvent-accessible, possibly aiding ATP/ADP release. The eBDIMS simulation suggests the nucleotide-binding site may be

affected during NACHT domain activation, possibly leading to ATP/ADP release. Notably, in the most recent activated NLRP3 structure (PDB ID: 8EJ4), ATP is present, suggesting a sequence of events: 1. ATP hydrolysis may first disrupt the closed state, 2. trigger conformational changes, 3. release ADP, and 4. enable additional ATP binding to stabilize the active NLRP3 inflammasome. We hope our findings may inspire further research on the ATP hydrolysis activity in the NLRP3 inflammasome activation and assembly.

A related point – It will be helpful if the authors discuss how the preparation of their NLRP3^{ΔPYD}/+ATP/-MCC950 sample differs from the previously solved NLRP3 cage, which may produce the novel open octamer structure?

And do they see any 2D/3D classes that may correspond to the closed NLRP3 cage in the NLRP3^{ΔPYD}/+ATP/-MCC950 dataset?

We purified NLRP3^{ΔPYD}/+ATP/-MCC950 using the same method as NLRP3^{ΔPYD}/+ATP/+MCC950, with the only difference being the absence of MCC950. Notably, without MCC950, the sample exhibited relatively rapid precipitation, occurring within 30 minutes on ice. Cryo-EM grids were prepared immediately after complex purification.

After initial 2D classification on the NLRP3^{ΔPYD}/+ATP/-MCC950 dataset (attached here), we observed significant particle heterogeneity, including small and large complexes. Following multiple rounds of 2D and 3D classifications, we identified a stable population of the open octamer complex but did not find the closed cage or closed hexamer.

Minor issues:

1. When discussing the residues involved in salt bridge pairs (page 4), a figure of side chain densities of corresponding amino acids should be included as a supplementary figure to validate the accuracy of the model-building process.

Thank you for the suggestion. We've added an additional figure in Supplementary Figure 3D to illustrate the density around residues possibly involved in salt bridge pairs. Furthermore, we've rephrased the sentence in the main text on page 4 to read as follows: "NBD- Lys133, Arg137, Lys138, Lys142, Lys289 can **potentially** form salts-bridges with corresponding LRR- Asp804, Asp807, Glu864, Glu1007, Asp750, respectively (Fig. 2C, Fig. S3D)."

2. Line 2 on page 4- It is less clear what the authors mean by "Superpositions of pairs of equivalent subdomains within NACHT further improve the structural similarities and highlight the plasticity within NLRP3."

We have rephrased the sentence in the main text as follows: "Superpositions of equivalent subdomains (NBD, HD1, WHD, HD2, and LRR) from the closed, open, and activated NLRP3 structures further improve the structural similarities, highlighting the NLRP3's plasticity with predominant subdomain rigid body movements during conformational changes (Fig. S4, and Table S1)."

Additionally, we include Supplementary Figure S4 and Table S1 (attached here) to clarify the similarities between equivalent subdomains during conformational changes.

Fig. S4. Subdomain structural comparisons. NLRP3 subdomains (NBD, HD1, WHD, HD2, and LRR) from the closed cage (in gray, PDB ID: 7PZC), open cage (color-coded as diagram above, this study), and activated NLRP3 inflammasome (in red, PDB ID: 8EJ4) were superimposed. Highlighted regions (helix₁₇₉₋₁₉₂, helix₅₈₆₋₅₉₆, and hairpin loop₅₀₈₋₅₁₅) from the activated NLRP3 are newly formed during inflammasome assembly, while the acidic loop₆₈₃₋₇₂₆ from the closed cage is also highlighted.

RMSD (Å)	NBD	HD1	WHD	HD2	LRR
Close cage vs NEK7 binding	2.31409	1.31659	1.59072	2.02322	2.07873
Close vs Open cages	2.42057	2.22592	2.26938	2.43403	2.0617
Activated vs Open cage	2.36169	1.52022	4.85262	2.76924	0.973654
Activated vs Close cage	2.94385	2.65039	4.5396	2.59787	1.67248

Table. S1. Average main chain RMSD values (in Å) between subdomains (NBD, HD1, WHD, HD2, and LRR) of NLRP3 from Close cage (PDB ID: 7PZC), NEK7 binding (this study), open cage (this study), and activated NLRP3 inflammasome (PDB ID: 8EJ4). The RMSD values were calculated using Superpose in CCP4.

REVIEWER COMMENTS

Reviewer #1 (Remarks to the Author):

The authors have addressed some of my comments, and the manuscript is also well prepared. However, as the authors mentioned in the rebuttal letter, they are not able to finish some experiments in the current manuscript. As a result, I am not totally convinced that the structure reported by this manuscript is a bona fide intermediate state of NLRP3 inflammasome.

Refer to:

In this study, the authors report the cryo-EM structure of PYD-truncated human NLRP3 in the form of an open octamer, which is different from the previously reported cage-shaped inactive form or disk-shaped active form. Considering the structural re-organization of NOD module in the open octamer, the authors propose this is the intermediate state during NLRP3 inflammasome activation and assembly. A 293T cell-based NLRP3 inflammasome activation assay is employed to show that mutations on the open octamer's interaction interface attenuate the IL-1 β secretion.

This is a very interesting body of work, however, there are several points that I would like the authors to consider.

Specific points:

1. Since the determined structure is from the PYD domain truncated NLRP3 protein, whether the full-length NLRP3 could also form the open octamer structure? How to prove that by additional structural or biochemical studies?
2. The authors mention the "self-activation of NLRP3" in the title and the manuscript, however, I can not understand clearly what does that mean. Do the authors mean the NLRP3 can be activated in the absence of any stimuli, or once activated, the NLRP3 could be activated in a self-propagation manner similar to NLRC4? Whatever, I can not find any evidence to support either conclusion in the current manuscript.
3. In the NLRP3 inflammasome activation assay performed in the 293T cell, the authors tried three different doses of transfected NLRP3 plasmid, and measured the IL-1 β secretion without any NLRP3 activation stimuli. Firstly, the ASC speck formation and caspase-1 activation need to be determined among different NLRP3 mutants; Secondly, what is the physiology concentration of NLRP3 in vivo, such as the BMDM cells, and a similar expression level of NLRP3 is expected to perform assay in 293T cells; Last but not least, the key NLRP3 mutations need to be verified in immune cells, such as BMDMs or THP1, and then performed the NLRP3 activation assay in a more physiological condition (LPS+ATP or LPS+Nigericin).
4. What is the relationship between the NEK7 binding and this open octamer formation? Does the NLRP3 expose more NEK7 binding surface in the open octamer compared with the inhibitory cage-shaped structure?
5. What is driving force for NLRP3 to form this open octamer structure from the cage-shaped structure? The hydrolysis of ATP? How the structural re-organization of NOD module leads to the change of oligomeric form of NLRP3?
6. The figure quality need to be improved. For example, what do two different colors mean in Fig. 1B? The whole picture of NLRP3 octamer shown in Fig. 1B should contain multiple side views to make it easier to understand. The label of "one unit" is not clear in Fig. 2A and it is a little hard for readers to understand.

Reviewer #2 (Remarks to the Author):

In their revision, the authors have adequately addressed my comments.

Reviewer #3 (Remarks to the Author):

The authors have addressed all my questions.

REVIEWER COMMENTS

Reviewer #1 (Remarks to the Author):

The authors have addressed some of my comments, and the manuscript is also well prepared. However, as the authors mentioned in the rebuttal letter, they are not able to finish some experiments in the current manuscript. As a result, I am not totally convinced that the structure reported by this manuscript is a bona fide intermediate state of NLRP3 inflammasome.

Thank you for the reviewer's feedback and recognition of the improvement in our manuscript. The authors acknowledge two major limitations in our study: firstly, the structure was determined using NLRP3 without the PYD domain, and secondly, the ASC speck formation and caspase-1 activation need to be determined among different NLRP3 mutants, and the findings were tested in HEK293T cells but not verified in human immune cells such as BMDMs or THP1.

As for the first limitation, studying the full-length WT NLRP3 posed challenges for structural analysis. The full-length WT NLRP3 was captured in either the closed form with MCC950 or the fully activated form with ASC-PYD, NEK7, ATP analogue (ATPγS), and nigericin treatment. Our open NLRP3ΔPYD without MCC950 captured an intermediate state, representing a transition from the closed cage to a fully activated inflammasome complex.

Regarding the second limitation, we are pleased to present new results on ASC formation. Utilizing an ASC-mCherry stable transduced HEK293T cell line, we captured the effects of mutations on ASC speck formations, outlined below:

“

Fig. 4. Interfaces in the open octamer are critical in the NLRP3 activation and assembly.

A. Mutations at the interfaces did not impair NLRP3 expression level. The component proteins were detected using Western blots. B. Cellular assay showing induction of IL-1 β signaling upon transfection with 2.5, 5, or 10 ng of WT or mutant FL NLRP3 vectors. Data points with P-values lower than 0.0001, 0.01, or 0.1 were denoted as four stars, two stars, or single star, respectively. These experiments were repeated twice. EV, empty vector control. C. Structural and sequence alignments around the catalytic surface around Asp363 (top) and the Loop504-517 region (lower), respectively. Closed (PDB ID: 7LFH), open (this study), activated (PDB ID: 8EJ4) NLRP3, NLRC4 (PDB ID: 8FW2), and NAIP (PDB ID: 8FVU) were shown as cartoon and in cyan, slate, light green, pink, and orange, respectively. The adjacent activated NLRP3 protomer was colored in dark green. Side chains of selected residues were shown in sticks in the structural alignments and highlighted with filled gray boxes in the sequence alignments. **D. Fluorescent mCherry measurements in HEK293TASC-mCherry cells transfected with empty vector (EV), NLRP3 WT, or mutants, with/without Nigericin treatment. Two cell images were shown as reference.**

In the main text, a new section was introduced:

“Oligomerization assists NLRP3 ASC speck formation, bypassed by Nigericin

We further explored the influence of oligomerization states on NLRP3 ASC speck formation^{28,29}. The mCherry-tagged ASC was stably transduced into HEK293T cells, which were then transiently transfected with different amounts (2.5 ng, 5 ng, and 10 ng) of human NLRP3 constructs carrying mutations in the oligomerization interfaces (Fig. 4D). Evaluation of mCherry signals was performed using the Incucyte S3 (2022B Rev2 software) with basic analyzer and surface-fit analysis. Without NLRP3, mCherry signals diffused within the cells. In contrast, in the presence of NLRP3 with or without Nigericin, we observed intense mCherry clusters, signaling ASC speck formation. Consistent with our IL-1 β signaling experiments after NLRP3 transfection, mutations in the oligomerization interfaces influenced ASC speck formations, while, the Back-Back interface mutation (F813D) enhanced ASC speck formation. Notably, Nigericin significantly enhanced ASC speck formation in both WT and mutated NLRP3 (Fig. 4D), along with heightened IL-1 β signaling (Fig. S8C). Our cellular results collectively highlight the critical role of oligomerization in NLRP3 activation and inflammasome assembly. Furthermore, Nigericin was observed to potentially bypass the oligomerization requirement, efficiently activating NLRP3 and inducing ASC speck formation.”

We recognize the need to perform NLRP3 activation assays under more physiological conditions (LPS+ATP or LPS+Nigericin) in human BMDMs or THP1. In the main text (page 14), we introduced: **“Our cellular studies were based on transiently transfecting NLRP3 mutants in HEK293T cells.** Yet, we can't rule out alternate pathways in NLRP3 inflammasome activation, like NLRC4 crosstalk, Nigericin, or lipopolysaccharide (LPS) activation in human macrophages²¹. **Our cellular assay demonstrated that Nigericin more effectively induced NLRP3 activation and subsequent IL-1 β signaling (Fig. 4D, S8C), implying that Nigericin may promote NLRP3 activation independently of oligomeric states. The authors hope that our present studies will serve as inspiration for future, advanced studies in human immune cells, such as BMDMs or THP1.**”

Note that our current results suggest Nigericin can directly activate NLRP3, bypassing oligomerization facilitation. This finding may inform future studies in human immune cells like BMDMs or THP1 with LPS+Nigericin.

Regarding other comments, the authors have addressed those points in our previous version. For a detailed summary, please refer to the attached response.

Refer to:

In this study, the authors report the cryo-EM structure of PYD-truncated human NLRP3 in the form of an open octamer, which is different from the previously reported cage-shaped inactive form or disk-shaped active form. Considering the structural re-organization of NOD module in the open octamer, the authors propose this is the intermediate state during NLRP3 inflammasome activation and assembly. A 293T cell-based NLRP3 inflammasome activation assay is employed to show that mutations on the open octamer's interaction interface attenuate the IL-1 β secretion.

This is a very interesting body of work, however, there are several points that I would like the authors to consider.

Specific points:

1. Since the determined structure is from the PYD domain truncated NLRP3 protein, whether the full-length NLRP3 could also form the open octamer structure? How to prove that by additional structural or biochemical studies?
2. The authors mention the “self-activation of NLRP3” in the title and the manuscript, however, I can not understand clearly what does that mean. Do the authors mean the NLRP3 can be activated in the absence of any stimuli, or once activated, the NLRP3 could be activated in a self-propagation manner similar to NLRC4? Whatever, I can not find any evidence to support either conclusion in the current manuscript.
3. In the NLRP3 inflammasome activation assay performed in the 293T cell, the authors tried three different doses of transfected NLRP3 plasmid, and measured the IL-1 β secretion without any NLRP3 activation stimuli. Firstly, the ASC speck formation and caspase-1 activation need to be determined among different NLRP3 mutants; Secondly, what is the physiology concentration of NLRP3 in vivo, such as the BMDM cells, and a similar expression level of NLRP3 is expected to perform assay in 293T cells; Last but not least, the key NLRP3 mutations need to be verified in immune cells, such as BMDMs or THP1, and then performed the NLRP3 activation assay in a more physiological condition (LPS+ATP or LPS+Nigericin).
4. What is the relationship between the NEK7 binding and this open octamer formation? Does the NLRP3 expose more NEK7 binding surface in the open octamer compared with the inhibitory cage-shaped structure?
5. What is driving force for NLRP3 to form this open octamer structure from the cage-shaped structure? The hydrolysis of ATP? How the structural re-organization of NOD module leads to the change of oligomeric form of NLRP3?
6. The figure quality need to be improved. For example, what do two different colors mean in Fig. 1B? The whole picture of NLRP3 octamer shown in Fig. 1B should contain multiple side views to make it easier to understand. The label of “one unit” is not clear in Fig. 2A and it is a little hard for readers to understand.

Reviewer #2 (Remarks to the Author):

In their revision, the authors have adequately addressed my comments.

Reviewer #3 (Remarks to the Author):

The authors have addressed all my questions.

We are grateful for the valuable feedback provided by Reviewers #2 and #3.

REVIEWERS' COMMENTS

Reviewer #1 (Remarks to the Author):

In their revision, the authors have addressed my concern.